# Differential contribution of TFE3 isoforms to cell motility and invasion

Pablo S Contreras [ID][1,2], José A Martina [ID][1,2], Katie Rollins [ID][1], Eutteum Jeong[1], Alberto Rissone[1] & Rosa Puertollano [ID][1][✉]

## Abstract

TFE3 orchestrates cellular responses to a variety of stress conditions, promoting restoration of cellular homeostasis and cell survival. Here we report the presence of two different TFE3 isoforms generated by the use of alternative transcription initiation sites. The long isoform (TFE3-L) undergoes continuous proteolytic degradation due to the presence of a phosphodegron in its N-terminal region and only accumulates under specific stress conditions. In contrast, the short isoform (TFE3-S) lacks the first 105 residues containing the phosphodegron and is constitutively expressed at high levels in most cell types. Both isoforms share the same Rags/mTORC1-dependent mechanism of regulation and display comparable capacity of inducing expression of lysosomal and autophagic genes upon activation. However, TFE3-L is considerably more efficient than TFE3-S promoting cell migration and invasion. Accordingly, specific TFE3-L depletion in a cellular model for tuberous sclerosis causes a significant reduction in cell motility and invasiveness. Our data reveal that the two TFE3 isoforms exhibit partial redundancy and that the appearance of TFE3-L following prolonged stress potentially correlates with metastatic behaviors.

**Keywords** TFE3; Lysosomes; Autophagy; Isoforms; Motility
**Subject Categories** Autophagy & Cell Death; Cancer; Organelles

## Introduction

TFEB and TFE3 are members of the MiT/TFE family of basic helix–loop–helix leucine zipper transcription factors known for orchestrating cellular responses to a variety of stress conditions, including nutrient deprivation, oxidative stress, pathogen infection, and organelle damage (La Spina et al, 2020). TFEB and TFE3 regulate expression of hundreds of genes, leading to induction of lysosomal biogenesis, autophagy, and metabolic reprograming, thus contributing to restoration of energy homeostasis and cell survival (Raben and Puertollano, 2016; Sardiello et al, 2009; Settembre et al, 2011). The key role played by these transcription factors is evidenced by their implication in multiple human pathologies, such as cancer, neurodegeneration, and immune diseases (Contreras and Puertollano, 2023).

The activity of TFEB and TFE3 is mainly regulated by the control of their intracellular distribution (Puertollano et al, 2018). Under basal conditions, TFEB and TFE3 are mostly excluded from the nucleus. Cytosolic retention requires the recruitment of the transcription factors to the lysosomal limiting membrane through interaction with Rag GTPases (Martina and Puertollano, 2013) and their subsequent phosphorylation by the serine/threonine kinase mTORC1 (Martina et al, 2012; Martina et al, 2014; Roczniak-Ferguson et al, 2012; Settembre et al, 2012). Phosphorylation of TFEB at serine 211 (S211) and TFE3 at S321 induces binding to 14-3-3 and prevents nuclear shuttling (Martina et al, 2012; Martina et al, 2014; Roczniak-Ferguson et al, 2012). Conversely, inactivation of mTORC1 in response to stress, together with the activation of specific phosphatases, inhibits TFEB/3 phosphorylation, promoting activation (Martina and Puertollano, 2018; Medina et al, 2015). The Folliculin (FLCN)/Folliculin-interacting protein (FNIP) complex plays also an important role in TFEB and TFE3 regulation. By promoting GDP-loading of RagC, FLCN facilitates interaction of TFEB/TFE3 with Rags, thus inducing cytosolic retention (Cui et al, 2023; Li et al, 2022; Napolitano et al, 2020). A growing number of studies have shown that pathological or stress conditions that block FLCN function, or in general the interaction of TFEB and TFE3 with Rags, will prevent the mTORC1-dependent phosphorylation of the transcription factors, even if the kinase itself remains active, causing their nuclear translocation (Alesi et al, 2021; Jeong et al, 2024; Malik et al, 2023; Nakamura et al, 2020; Tapia et al, 2025; Xu et al, 2025). Finally, a wide range of additional post-translational modifications provide further fine-tuned regulation by controlling protein stability, nuclear export, and DNA binding affinity (Takla et al, 2023).

We and others have previously reported that TFE3 migrates as two distinct bands of approximately 72 and 82 kDa in SDS-PAGE (Martina et al, 2014). Interestingly, whereas the smaller form of TFE3 is always present in cell lysates, the higher molecular weight form appears only after specific stress conditions, such as prolonged starvation (Martina et al, 2014). A recent report suggested that the 82 kDa form does in fact corresponds to TFE3 full length (Nardone et al, 2023). The presence of a phosphodegron in the N-terminal portion of TFE3 causes its continuous

[1]Cell and Developmental Biology Center, National Heart, Lung, and Blood Institute, National Institutes of Health, Bethesda, MD, USA. [2]These authors contributed equally: Pablo S Contreras, José A Martina. ✉E-mail: puertolr@mail.nih.gov

degradation in basal conditions. Phosphorylation of TFE3 at serine 47 (S47) induces CUL1$^{\beta\text{-TrCP1/2}}$-mediated ubiquitination and subsequent proteasomal degradation, thus suppressing TFE3 activity. Mutation of S47, or prevention of its phosphorylation under stress condition, is sufficient to stabilize TFE3, facilitating expression of target genes (Nardone et al, 2023). It was proposed that the control of TFE3 stability may constitute the main mechanism of TFE3 regulation.

There is, however, a critical question that remains to be addressed regarding the role of the 72 kDa TFE3 short form (TFE3-S). TFE3-S is expressed at high levels in most cell types and its expression remains constant both under control and stress conditions. Characterizing the nature and regulation of TFE3-S, as well as its potential transcriptional activity and selectivity is, therefore, essential to fully understand TFE3 regulation.

## Results

### Expression of TFE3 long and short forms in response to different stress conditions

To better understand the relationship between TFE3 long (TFE3-L) and short (TFE3-S) forms, we subjected ARPE19 cells to starvation for different periods of time. In agreement with our previous observations (Martina et al, 2014), we found that the levels of TFE3-L were low under basal conditions but progressively increased following prolonged starvation (Fig. 1A,B; Appendix Fig. S1A). In contrast, TFE3-S levels remained largely constant under control or stress conditions (Fig. 1A). Both forms showed a small decrease in electrophoretic mobility under starvation, which is indicative of reduced mTORC1-dependent phosphorylation (Fig. 1A) (Martina et al, 2014), and TFE3 nuclear accumulation was observed as early as 1 h following incubation with EBSS (Fig. EV1A). Similar results were observed in HEK293T and HeLa cells (Fig. EV1B–D; Appendix Fig. S1B,C). When cells were placed back in complete medium after starvation (refed), we observed a rapid mTORC1 reactivation that was followed by a decrease in TFE3-L levels (Fig. 1C,D). TFE3-L degradation during refed was inhibited by Torin-1 (Appendix Fig. S2A–C), which is consistent with the suggestion that mTORC1-dependent phosphorylation of S47 activates TFE3 phosphodegron (Nardone et al, 2023), as well as by incubation with the proteasome inhibitor MG132 (Fig. 1E,F).

We and others have shown that interaction of TFEB and TFE3 with active Rags GTPases is required for recruitment of the transcription factors to the lysosome surface and subsequent mTORC1 phosphorylation (Martina et al, 2014; Martina and Puertollano, 2013). Therefore, we predicted that inhibiting TFE3 interaction with Rags should result in TFE3-L stabilization. In agreement with this idea, we observed that expression of inactive Rags was sufficient to significantly increase TFE3-L levels, both in ARPE19 and HEK293T cells (Fig. EV1E–H). Likewise, depletion of folliculin (FLCN), a component of the FNIP/FLCN GAP complex that promotes GDP-loading of RAGC (Lawrence et al, 2019), resulted in TFE3-L stabilization (Fig. 1G–I). In agreement with previous reports (Nardone et al, 2023), we also confirmed that inhibition of Cullin-RING E3 ligases with MLN4924 prevented TFE3-L degradation, resulting in its accumulation (Figs. 1J,K and EV1E,F).

TFE3 is activated by a wide variety of stress conditions (Raben and Puertollano, 2016). Accordingly, we observed TFE3 translocation to the nucleus in RAW 264.7 macrophages treated with LPS (Fig. EV1I) (Pastore et al, 2016). However, LPS treatment did not increase TFE3-L levels (Fig. EV1J), suggesting that TFE3-S can be activated independently of the presence of TFE3-L, and that the short form may play a predominant role in response to certain types of stress.

To further corroborate this observation, we incubated ARPE19 cells with different stressors, including inductors of oxidative stress (NaAsO$_2$), mitochondrial damage (CCCP), lysosomal rupture (LLOMe), lysosomal alkalinization (Chloroquine), starvation (EBSS), and mTOR inhibition (Torin-1). At short incubation times (2 h and 4 h), all the stressors, except for CCCP, induced strong TFE3 activation, as assessed by the dephosphorylation of S321, the residue that mediates binding to 14-3-3 and consequent cytoplasmic retention, as well as by the accumulation of TFE3 in the nucleus (Fig. 1L,M). However, we did not observe a robust TFE3-L increase at these early times (Fig. 1L), indicating that prolonged stress is required to accumulate significant levels of TFE3-L. These results also suggest that TFE3-S may play a prevalent role in early stress responses. Consistently, subcellular fractionation analysis of ARPE19 cells treated with NaAsO$_2$, LLOMe, or EBSS for 2 h confirmed that most of the TFE3 present in nuclear fractions at this time corresponds to TFE3-S (Appendix Fig. S3A–C). A significant TFE3-L accumulation was, however, observed following prolonged incubation (12 h and 24 h) with all the tested stressors (Appendix Fig. S4A–C). The only exception was LLOMe, which induced efficient TFE3 activation at 12 h but not at 24 h, a phenomenon that may be indicative of the ability of cells to eliminate or repair damaged lysosomes over time. Altogether, our data suggest that the relative contribution of the two TFE3 forms may vary depending on the type and duration of the stress.

### TFE3-L and TFE3-S are generated by the use of alternative transcription initiation sites

Several functional domains have been described in TFE3 (La Spina et al, 2020), including a Rag-binding domain (RBD), an activation domain (AD), a 14-3-3 binding motif, a basic helix-loop-helix domain (bHLH) that mediates DNA binding, and a leucine-zipper domain (LZ) implicated in homo- and heterodimerization (Fig. EV2A). Since it has been proposed that TFE3-L corresponds to the full-length protein, an important question is determining which domains are present in the TFE3-S and how this isoform originates. The analysis of genetic databases shows that TFE3 gene is composed of at least ten coding exons with a start codon in exon 1; however, a potential second start codon is present in exon 3, which might produce a shorter variant (Fig. EV2A). To investigate the presence of TFE3-S transcripts, we performed reverse-transcription PCR analysis on RNA extracted from ARPE19 cells, using a common reverse primer targeting exon 4 in combination with different forward primers to specifically target the 5'UTR of TFE3-L or the intronic region right upstream exon 3 (Fig. EV2B). As expected, we observed two PCR products (a and b) corresponding to the predicted TFE3-L transcripts. However, we also detected two specific PCR fragments (c and d) corresponding to the proposed TFE3-S transcripts (Fig. EV2B). To further corroborate the purity of the cDNA preparation, we performed

  

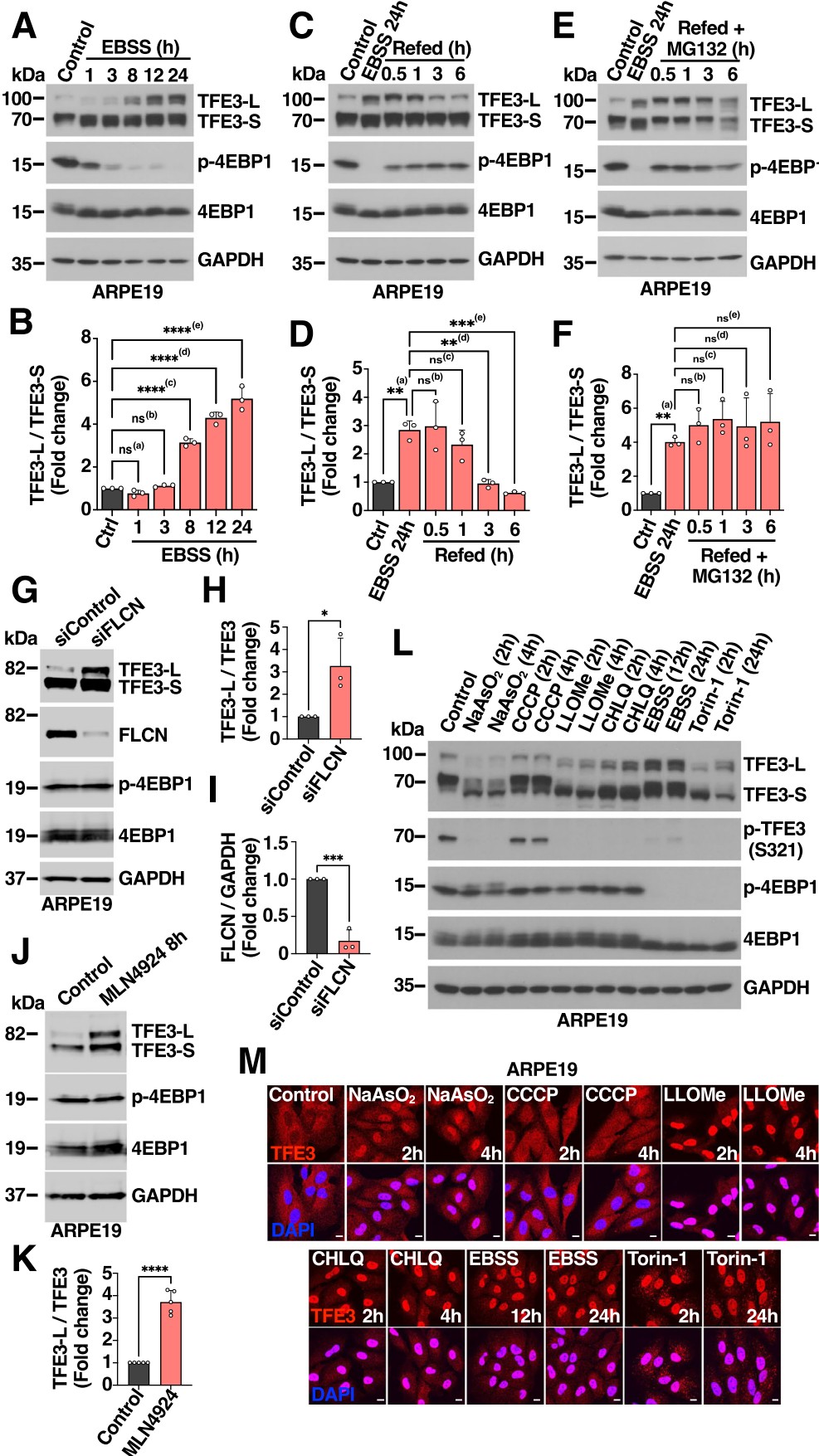

**Figure 1. Differential expression of TFE3 isoforms in response to stress.**

(A) Immunoblot analysis of protein lysates from ARPE19 cells treated with EBSS for 1, 3, 8, 12, and 24 h. (B) Quantification of protein levels showing TFE3-L/TFE3-S ratio expressed as fold change as shown in (A). Data are presented as mean ± SD of three independent experiments. (ns) not significant [(a)]$P = 0.7522$; (ns) not significant [(b)]$P = 0.9678$; [****(c)]$P < 0.0001$; [****(d)]$P < 0.0001$; [****(e)]$P < 0.0001$ (one-way ANOVA followed by Dunnett's multiple comparison post-test). (C) Immunoblot analysis of protein lysates from ARPE19 cells treated with EBSS for 24 h followed by incubation in complete media (Refed) for 0.5, 1, 3, and 6 h. (D) Quantification of protein levels showing TFE3-L/TFE3-S ratio expressed as fold change as shown in (C). Data are presented as mean ± SD of three independent experiments. [**(a)]$P = 0.0023$; (ns) not significant [(b)]$P = 0.9987$; (ns) not significant [(c)]$P = 0.6947$; [**(d)]$P = 0.0019$; [***(e)]$P = 0.0005$ (one-way ANOVA followed by Tukey's multiple comparison post-test). (E) Immunoblot analysis of protein lysates from ARPE19 cells treated with EBSS for 24 h before the addition of complete media (Refed) in the presence of the proteosome inhibitor MG132 for 0.5, 1, 3, and 6 h. (F) Quantification of protein levels showing TFE3-L/TFE3-S ratio expressed as fold change as shown in (E). Data are presented as mean ± SD of three independent experiments. [**(a)]$P = 0.0069$; (ns) not significant [(b)]$P > 0.9999$; (ns) not significant [(c)]$P = 0.5996$; (ns) not significant [(d)]$P > 0.9999$; not significant [(e)]$P > 0.9999$ (one-way ANOVA followed by Newman–Keuls multiple comparison post-test). (G) Immunoblot analysis of protein lysates from ARPE19 cells treated with siRNA against FLCN for 72 h. (H) Quantification of protein levels showing TFE3-L/TFE3 ratio expressed as fold change as shown in (G). Data are presented as mean ± SD of three independent experiments. [*]$P = 0.0343$ (unpaired Student's $t$ test). (I) Quantification of protein levels showing FLCN/GAPDH ratio expressed as fold change as shown in (G). Data are presented as mean ± SD of three independent experiments. [***]$P = 0.0006$ (unpaired Student's $t$ test). (J) Immunoblot analysis of protein lysates from ARPE19 cells treated with 1 μM MLN4924 for 8 h. (K) Quantification of protein levels showing TFE3-L/TFE3 ratio expressed as fold change as shown in (J). Data are presented as mean ± SD of three independent experiments. [****]$P < 0.0001$ (unpaired Student's $t$ test). (L) Immunoblot analysis of protein lysates from ARPE19 cells treated with either 100 μM NaAsO$_2$, 10 μM CCCP, 1 mM LLOMe, 50 μM Chloroquine, EBSS, or 250 nM Torin-1 for 2, 4, 12 or 24 h. (M) Immunofluorescence confocal microscopy of ARPE19 cells showing the subcellular distribution of TFE3 (red) in response to treatments with either 100 μM NaAsO$_2$, 10 μM CCCP, 1 mM LLOMe, 50 μM Chloroquine, EBSS or 250 nM Torin-1 for 2, 4, 12, or 24 h. Scale bars: 10 μm. Source data are available online for this figure.

PCR analysis to assess the presence of genomic DNA (gDNA) contamination in the mRNA preparation. As expected, the cDNA sample lacked the ~1.2 kb band specific to the gDNA sample (Fig. EV2C). Finally, sequencing of the specific bands from samples (a) and (c) (~894 and ~597 bp, respectively) confirmed that they corresponded to the expected fragments (Fig. EV2D). Altogether, our data indicate that TFE3-L and TFE3-S represent the expression products of two distinct TFE3 transcripts, likely under the control of two independent transcription initiation sites (Fig. 2A).

The TFE3 RBD comprise two alpha helices that include the amino acids located between positions 112 and 206 (Martina et al, 2014; Nardone et al, 2023). According to our results, both the RBD and 14-3-3 binding motifs, which are required for cytosolic sequestration, should be present in TFE3-S (Fig. 2A). Accordingly, we found that TFE3-S is mostly located to the cytosol under control conditions and only translocated to the nucleus following mTORC1 inactivation with Torin-1 (Fig. 2B). It is also important to note that even though MLN4924 increased TFE3-L levels, the protein remained in the cytosol under basal conditions, indicating that Rag- and 14-3-3-mediated retention is also critical for TFE3-L regulation (Fig. 2B). To further confirm this, we performed pull-down experiments with active and inactive Rags. As expected, we detected interaction of both TFE3 isoforms with active Rags (Appendix Fig. S5A). In addition, phosphorylation of S321 (S216 in TFE3-S), the residue implicated in binding to 14-3-3, was significantly reduced in both isoforms by Torin-1 treatment (Appendix Fig. S5C).

Next, we aimed to generate recombinant versions of TFE3-L and TFE3-S. For this we produced several TFE3 mutants carrying a C-terminal Myc epitope tag (Fig. 2C). As expected, we confirmed that mutation of S47 prevented TFE3 degradation, as the protein levels of TFE3-S47A were much higher than those of TFE3-WT (Fig. 2D). We proposed that TFE3-S47A-Myc parallels endogenous TFE3-L and henceforward this mutant was referred to as recombinant TFE3-L or rTFE3-L. Likewise, a TFE3 mutant lacking the first 105 residues was considered equivalent to endogenous TFE3-S and was labeled rTFE3-S. Interestingly, mutation of rTFE3 first methionine (rTFE3-M1A) resulted in expression of a stable, lower molecular weight band that runs with the same

electrophoretic mobility as rTFE3-S, suggesting that translation was initiated at an internal methionine codon, most likely M106. The rTFE3-M1A mutant was recognized by the Myc antibody, but not by an antibody directed against the N-terminal portion of the protein (Fig. 2D). We have previously described that treatment with Torin-1 causes Rag-dependent recruitment of TFEB and TFE3 to lysosomes (Martina et al, 2014; Martina and Puertollano, 2013). As seen in Fig. 2E and Appendix Fig. S5B, both rTFE3-L and rTFE3-M1A were recruited to lysosomes after incubation with Torin-1, further corroborating the presence of the RBD in both proteins. Finally, the lack of effect of the MLN4924 inhibitor on rTFE3-M1A is consistent with the absence of the N-terminal phosphodegron (Fig. 2F). Altogether our data suggests that at least under certain in vitro conditions, the short isoform might be also generated by the use of the internal methionine 106 as an alternative translation initiation site. For this reason, we generated an additional mutant in which S47 and M106 were mutated to alanine (rTFE3-L-M106A) that was used as an additional control (Fig. 2C,D).

To further characterize the mechanism of activation of the TFE3 mutants, we performed immunofluorescence analysis. As expected, deletion of the first 105 residues, did not affect the canonical TFE3 regulation. rTFE3-L, rTFE3-S, and rTFE3-L-M106A remained retained in the cytosol under control conditions, while rapidly moving to the nucleus following starvation (Fig. 2G) or Torin-1 treatment (Appendix Fig. S5B). In summary, we confirmed that both rTFE3-L and rTFE3-S isoforms share the same mechanism of regulation, in which mTORC1, Rags, and 14-3-3 collaborate to keep the transcription factors retained in the cytosol in the absence of stress.

## TFE3-L and TFE3-S induce expression of lysosomal and autophagic genes with comparable efficiency

We next seek to compare the transcriptional capability of TFE3-L and TFE3-S. For this we generated adenovirus expressing either rTFE3-L or rTFE3-S and performed RNA-seq analysis of ARPE19 cells infected with Ad-Null, Ad-rTFE3-L, or Ad-rTFE3-S for 30 h (Datasets EV1 and EV2). The comparative principal component

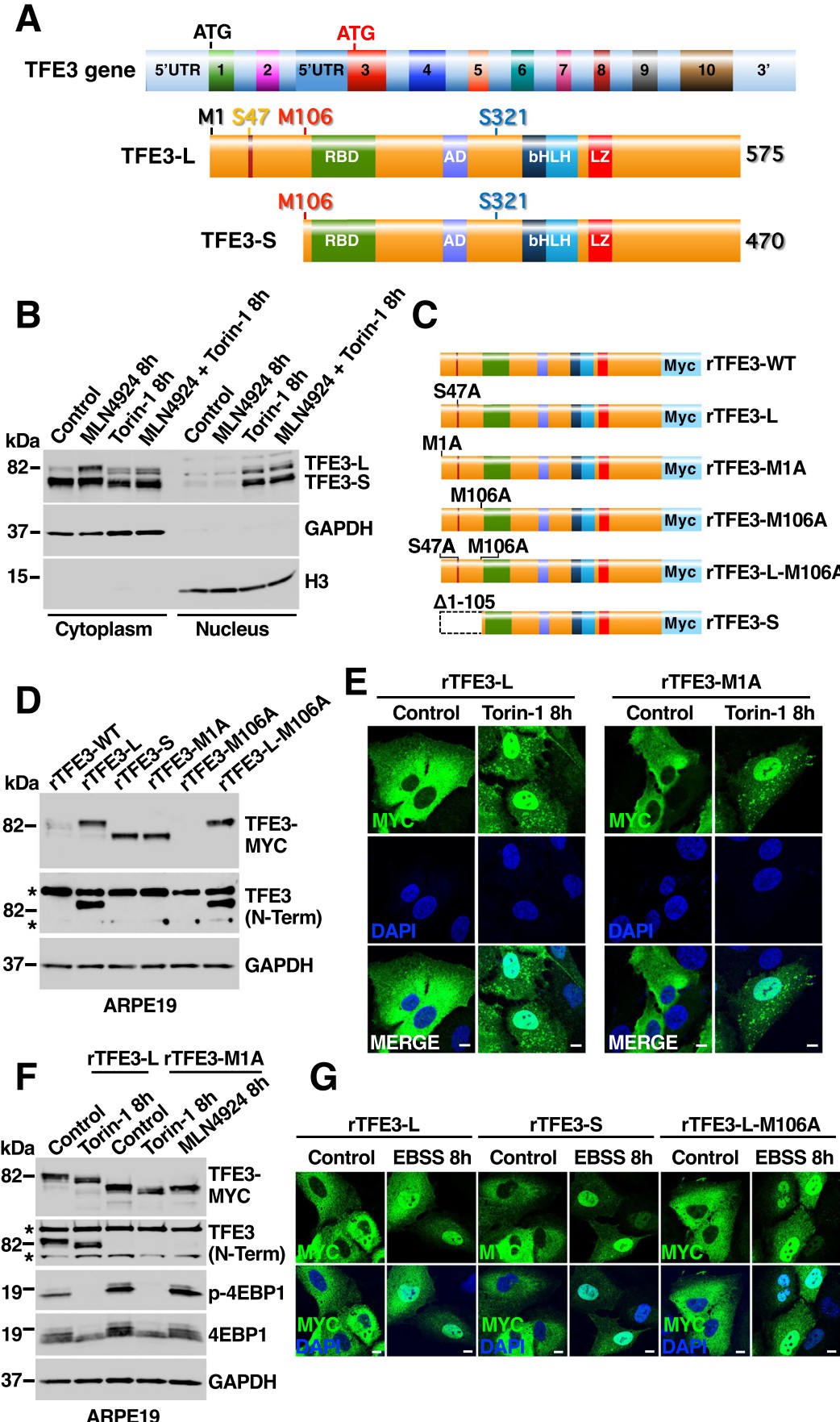

◀ **Figure 2. TFE3-L and TFE3-S are produced from alternative promoters and share the same mechanism of mTORC1 regulation.**

(A) Schematic diagram of the genomic structure of TFE3 representing exons 1 through 10, their corresponding intronic regions, the position of two start codons, and the conserved functional domains of TFE3-L and TFE3-S proteins. Intron sizes are not to scale, and exons and protein domains sizes and boundaries are approximate. (B) Immunoblot analysis of protein lysates from ARPE19 cells treated with either 1 μM MLN4924 or 250 nM Torin-1, or both together, for 8 h and subjected to subcellular fractionation. (C) Schematic diagram of recombinant human TFE3 proteins c-terminal tagged with a triple MYC showing the different introduced amino acid replacements and deletion. (D) Immunoblot analysis of protein lysates from ARPE19 cells transiently expressing the TFE3 proteins shown in (C) for 24 h. Asterisk (*) indicate unspecific band. (E) Immunofluorescence confocal microscopy of ARPE19 cells showing the subcellular distribution of recombinants TFE3-L-Myc or TFE3-M1A-Myc (green) in response to treatment with 250 nM Torin-1 for 8 h. Scale bars: 10 μm. (F) Immunoblot analysis of protein lysates from ARPE19 cells transiently expressing recombinant TFE3-L-Myc or TFE3-M1A-Myc and treated with either 250 nM Torin-1 or 1 μM MLN4924 for 8 h. Immunoblots are representative of at least three independent experiments. Asterisks (*) indicate unspecific bands. (G) Immunofluorescence confocal microscopy of ARPE19 cells showing the subcellular distribution of recombinants TFE3-L-Myc, TFE3-S and TFE3-M106A-Myc (green) in response to treatment with EBSS for 8 h. Scale bars: 10 μm. Source data are available online for this figure.

analysis between the three conditions showed that expression of either rTFE3-L or rTFE3-S induced profound changes in the transcriptional landscape when compared with control cells, while minor but still evident differences were observed when comparing rTFE3-L with rTFE3-S (Fig. 3A).

TFE3 is a well-recognized master regulator of lysosomal biogenesis. By simultaneously upregulating expression of multiple lysosomal genes, TFE3 increases both the activity and number of lysosomes (Martina et al, 2014). It was, therefore, not surprising that Cellular Compartment ontology analysis of those genes most upregulated by rTFE3-L (shrunkenLog2FC > 1, P-adjusted value < 0.001) revealed a clear enrichment in several classes linked to lysosomes (Fig. 3B). In agreement with previous studies, we also observed an over-representation of components associated with endosomes, mitochondria, and melanosomes (Fig. 3B) (Martina et al, 2016; Martina et al, 2014). Lysosomal-related categories were also the most significantly upregulated upon overexpression of rTFE3-S (Fig. 3C), and both isoforms increased lysosomal gene expression with similar efficiency (Fig. 3D). To corroborate our RNA-seq data, we analyzed expression of several lysosomal genes by RT-qPCR, including those encoding for components of the v-ATPase (ATP6V1C1 and ATP6V0A1), lysosomal transmembrane proteins (LAMP1, CTSN, and MCOLN1), luminal hydrolases (GLA and CTSA), and peripherally associated proteins (RRAGC). Comparative analysis of ARPE19 infected with Ad-Null, Ad-rTFE3-L, Ad-rTFE3-S, and Ad-rTFE3-L-M106A confirmed that all the mutants induced a robust and significant increase in expression of lysosomal genes (Fig. EV3A). TFE3 can also regulate expression of key regulators of the autophagy pathway to synchronize the cell degradative potential to increased energy demands (Martina et al, 2014). As seen in Fig. 3E, rTFE3-S was as efficient, and in some cases even more efficient than rTFE3-L, in promoting expression of autophagic genes, including UVRAG, WIPI1, ATG4A, ATG9B, MAP1LC3B, WDR81, and GABARAPL1.

To ensure that the high level of expression achieved by adenovirus infection was not masking some potential degree of selectivity between TFE3 isoforms, we transfected ARPE19 cells with plasmids encoding rTFE3-L and rTFE3-M1A for 12 h and monitored their ability to induce gene expression in response to stress. The level of protein expression was not too high under these conditions, as shown by the mainly cytosolic distribution of these mutants under control conditions (Fig. 2E). As expected, we found that both rTFE3-L and rTFE3-M1A induced a significant transcriptional upregulation of several well-known TFE3 targets in response to starvation and mTORC1 inhibition (Fig. EV3B).

Altogether, our results suggest that rTFE3-S, which is constitutively expressed in most cell types at high levels, has the same capability that rTFE3-L to induce lysosomal biogenesis and autophagy in response to stress.

## Overexpression of TFE3 isoforms leads to proliferation arrest, alterations in cell–cell contact, and increased migration

We next investigated which genes categories are downregulated by TFE3 overexpression. Ontology analysis (shrunkenLog2FC <-1, P-adjusted value <0.01) by Cellular Compartment (Fig. 4A) and KEGG pathways (Fig. EV4A) showed that infection of ARPE19 cells with either, Ad-rTFE3-L or Ad-rTFE3-S, caused a significant reduction in the levels of genes implicated in cell-cycle, cell–cell contact, and cytoskeleton regulation. To confirm these data, we first measured proliferation by incubating cells with 5-ethynyl-2'-deoxyuridine (EdU) followed by flow cytometry. As shown in Fig. EV4B,C, EdU incorporation analysis revealed over 80% reduction in the number of proliferative cells upon rTFE3-L or rTFE3-S expression. The Cip/Kip family of cyclin-dependent kinase inhibitors, which includes p27 Kip1, p57 Kip2 and p21 Waf1/Cip1, is a critical regulator of cell cycle progression. Cip/Kip family members bind and inhibit several cell cycle-promoting Cyclin/CDK complexes, enforcing G1/S restrictions, and over-expression of p21, p27, p57 halts cell cycle in multiple cell types (Csergeova et al, 2024). A previous study reported that TFEB overexpression causes p21 upregulation and consequent cell cycle inhibition (Pisonero-Vaquero et al, 2020). Therefore, we addressed the ability of TFE3 to modulate the levels of the Cip/Kip family members. Overexpression of either rTFE3-L or rTFE3-S induced a significant increase in p57 mRNA levels (Fig. EV4D), as well as p21 protein levels (Fig. EV4E,F). Inhibition of Cyclin/CDK complexes by p21 results in decreased phosphorylation of the retinoblastoma protein (Rb) and consequent inactivation of E2F-regulated genes (Niculescu et al, 1998; Xiong et al, 1993). Accordingly, we found that rTFEB3-L and rTFE3-S strongly decreased Rb phosphorylation when compared with control cells (Fig. EV4G–I). These data indicate that TFE3 inhibits cell proliferation when expressed at high levels, but the two isoforms do so with comparable efficiency.

Our RNA-seq analysis also revealed that overexpression of TFE3 isoforms caused a severe decrease in the expression of multiple genes implicated in cell–cell junction (Fig. 4B). These results were unexpected, since a role of TFE3 in cellular adhesion and tissue integrity has not been previously reported. One of the genes that appeared most significantly downregulated was JUP. This gene encodes gamma-catenin, a component of adherent junctions and

 

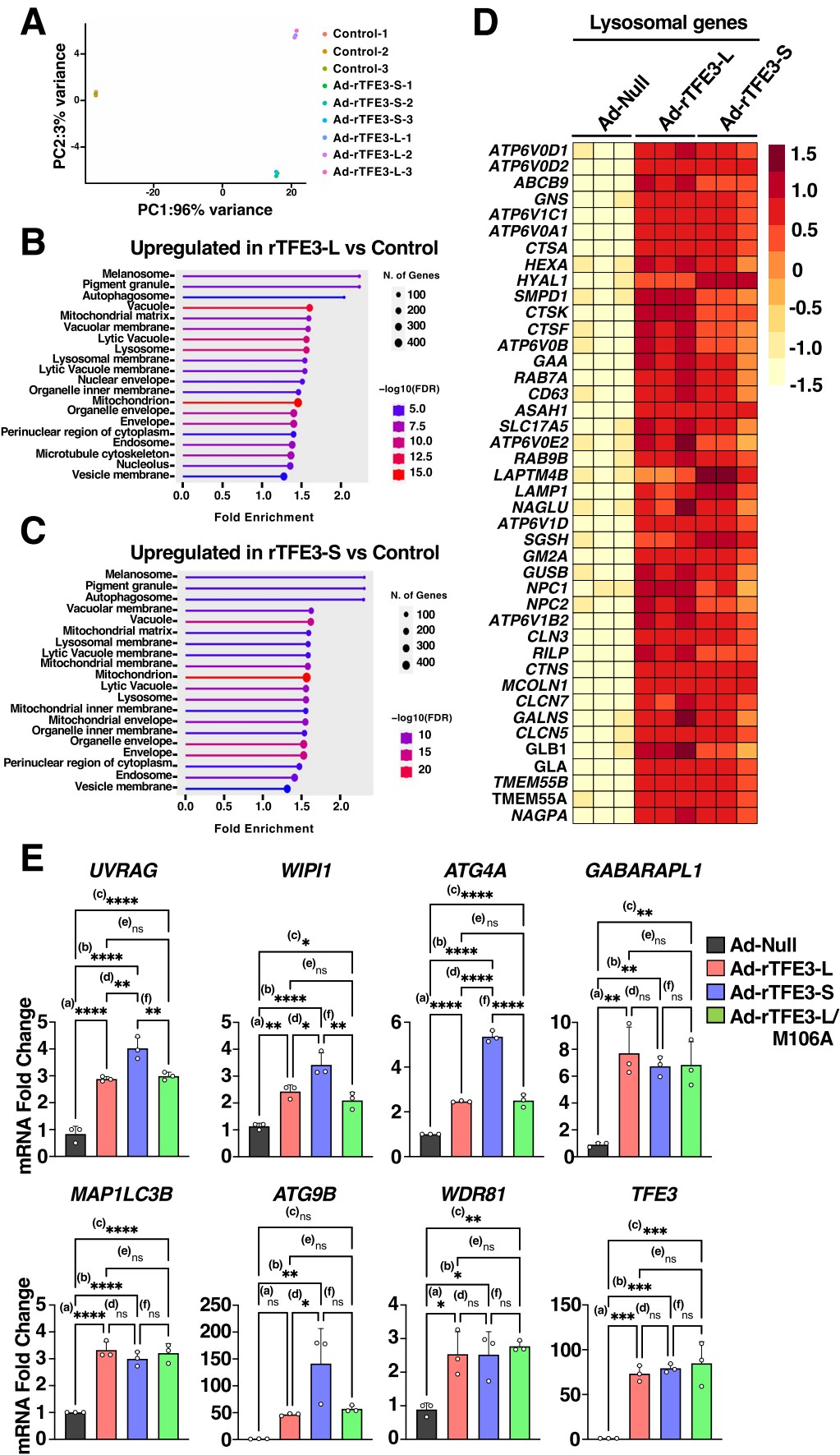

**Figure 3. TFE3-L and TFE3-S upregulate expression of lysosomal biogenesis and autophagy genes in response to stress.**

(A) Principal-component analysis (PCA) of genes with q-value < 0.05 shows distinct clustering of ARPE19 cells infected with adenovirus Control or adenovirus expressing recombinant TFE3-S-Myc or TFE3-L-Myc for 30 h. (B, C) KEGG pathway enrichment analysis of differentially expressed genes in ARPE19 cells infected with adenovirus expressing recombinant TFE3-L-Myc (B) or TFE3-S-Myc (C) for 30 h. Circle size represents the number of genes, while the color scale indicates the −log10 of the false discovery rate (FDR). (D) Heatmap of differentially expressed lysosomal genes in ARPE19 cells infected with adenovirus expressing recombinant TFE3-L-Myc, TFE3-S-Myc or Control (Ad-Null) for 30 h. (E) Relative quantitative RT-PCR analysis of the mRNA expression of autophagic genes (*UVRAG, WIPI1, ATG4A, GABARAPL1, MAP1LC3B, ATG9B* and *WDR81*) and TFE3 in ARPE19 cells infected with the indicated adenovirus for 30 h. Data are presented as mean ± SD of three independent experiments. UVRAG ($^{****(a)}P < 0.0001$; $^{****(b)}P < 0.0001$; $^{****(c)}P < 0.0001$; $^{**(d)}P = 0.0035$; (ns) not significant $^{(e)}P = 0.9596$; $^{**(f)}P = 0.0064$), WIPI1 ($^{**(a)}P = 0.0036$; $^{****(b)}P < 0.0001$; $^{*(c)}P = 0.0198$; $^{*(d)}P = 0.0171$; (ns) not significant $^{(e)}P = 0.5697$; $^{**(f)}P = 0.0032$), ATG4A ($^{****(a)}P < 0.0001$; $^{****(b)}P < 0.0001$; $^{****(c)}P < 0.0001$; $^{****(d)}P < 0.0001$; (ns) not significant $^{(e)}P = 0.9927$; $^{**(f)}P < 0.0001$), GABARAPL1 ($^{**(a)}P = 0.0012$; $^{**(b)}P = 0.0033$; $^{**(c)}P = 0.0029$; (ns) not significant $^{(d)}P = 0.8158$; (ns) not significant $^{(e)}P = 0.8574$, (ns) not significant $^{(f)}P = 0.9997$), MAP1LC3B ($^{****(a)}P < 0.0001$; $^{****(b)}P < 0.0001$; $^{****(c)}P < 0.0001$; (ns) not significant $^{(d)}P = 0.4872$; (ns) not significant $^{(e)}P = 0.9537$; (ns) not significant $^{(f)}P = 0.7678$), ATG9B ((ns) not significant $^{(a)}P = 0.3845$; $^{**(b)}P = 0.0035$; (ns) not significant $^{(c)}P = 0.2344$; $^{*(d)}P = 0.0313$; (ns) not significant $^{(e)}P = 0.9778$; (ns) not significant $^{(f)}P = 0.0544$), WDR81 ($^{*(a)}P = 0.0151$; $^{*(b)}P = 0.0158$; $^{**(c)}P = 0.0071$; (ns) not significant $^{(d)}P > 0.9999$; (ns) not significant $^{(e)}P = 0.9335$; (ns) not significant $^{(f)}P = 0.9229$), TFE3 ($^{***(a)}P = 00.0006$; $^{***(b)}P = 0.0004$; $^{***(c)}P = 0.0002$; (ns) not significant $^{(d)}P = 0.941$; (ns) not significant $^{(e)}P = 0.7137$; (ns) not significant $^{(f)}P = 0.9535$) (one-way ANOVA followed by Tukey's multiple comparison post-test). Source data are available online for this figure.

desmosomes that plays a critical role in cell adhesion. Infection of ARPE19 cells with either Ad-rTFE3-L or Ad-rTFE3-S resulted in a striking reduction in gamma-catenin protein and mRNA levels (Figs. 4C and EV4J). Furthermore, immunofluorescence analysis of ARPE19 cells stained with antibodies against beta-catenin further showed disruption of cell–cell contacts upon expression of TFE3 isoforms (Fig. EV4K).

Since defects in cell adhesion caused by the loss of gamma-catenin results in increased migration, and abnormal gamma-catenin function has been involved in tumor progression and metastasis (Sechler et al, 2015; Xia et al, 2024), we next measured the potential effect of TFE3 overexpression in cell migration. For this we performed scratch wound healing assays in ARPE19 cells infected with Ad-Null, Ad-rTFE3-L or Ad-rTFE3-S (end point after 48 h). Remarkably, we found that although expression of rTFE3-S modestly increased migration when compared with control cells, rTFE3-L was much more efficient increasing motility (Fig. 4D,E). The migration, however, was very disorganized; cells expressing rTFE3-L appeared elongated and randomly oriented, and often showed axon-like protrusions, similar to the ones described in certain metastatic cells (Yang et al, 2019) (Movies EV1–3 and arrowheads in Fig. 4D). Increased migration in response to TFE3-L, but not TFE3-S, expression was also observed in HeLa cells (Appendix Fig. S6). To further examine the potential contribution of TFE3 to metastasis, we carried out Boyden chamber invasion assays. In agreement with our motility experiments, we found that rTFE3-L, but not rTFE3-S, significantly increased the invasiveness of ARPE19 cells (Fig. 4F,G). This indicates that the TFE3 long isoform may specifically correlate with metastatic and invasive behaviors.

Autophagy-mediated degradation of focal adhesions has a profound effect on cell migration and invasion (Bressan et al, 2020; Kenific et al, 2016; Lu et al, 2021). To better understand the underlying mechanism of TFE3-L-induced migration, we assessed the effect of TFE3-L overexpression in control and ATG7-depleted cells. Efficient ATG7 depletion, as well as comparable TFE3-L expression, was monitored by qPCR and immunoblot (Appendix Fig. S7A–C). We found that the ability of TFE3-L to significantly reduce γ-catenin expression and increase invasion was not affected by depletion of ATG7 (Appendix Fig. S7D–G), suggesting that the effect of TFE3-L on cell–cell adhesion and migration is not just

mediated by its ability to induce autophagy. In contrast, the invasive capacity of TFE3-L was significantly reduced by mutation of its nuclear import signal, highlighting the requirement of TFE3-L transcriptional activity (Appendix Fig. S8A–C).

## TFE3-L and TFE3-S show selectivity for specific gene targets

Our migration experiments suggested that rTFE3-L and rTFE3-S may not share completely redundant functions and that the appearance of the long isoform following prolonged stress may confer cells with distinctive additional properties. To test this hypothesis, we searched our RNA-seq data for genes that showed differential expression between the two isoforms (Fig. 5A). Among the genes that were preferentially upregulated by rTFE3-L, we found several implicated in epithelial-to-mesenchymal transition (EMT), a process in which epithelial cells lose adhesion properties while acquiring mesenchymal features, such as invasiveness and increased motility. These included the integrin *ITGAX* (also known as CD11c), the metalloenzyme *ACP5*, the glycoprotein *CHI3L1* (YKL-40), and the long non-coding RNA *SNHG15*, all of which have been linked to metastasis, increased tumor aggressiveness, and poor prognosis (Chao et al, 2010; Geng et al, 2018; Hao et al, 2021; Hu et al, 2024; Morera et al, 2019; Reithmeier et al, 2017; Shen et al, 2023; Zhang et al, 2023). To validate our RNA-seq data, we performed RT-qPCR analysis. As seen in Fig. 5B, rTFE3-L and rTFE3-L-M106A caused a very robust increase in the expression of these four genes. Conversely, rTFE3-S failed to induce a significant upregulation despite comparable expression levels between the TFE3 isoforms. This unexpected gene expression selectivity was further confirmed by immunoblot and immunofluorescence. Whereas CD11c and YKL-40 protein levels were negligible in control and rTFE3-S expressing ARPE19 cells, infection with adenovirus encoding either rTFE3-L or rTFE3-L-M106A was sufficient to induce a robust upregulation (Fig. 5C–E).

We also observed differential expression of several critical transcriptional regulators. Particularly intriguing were *CITED1* and *CEBPA*, two well-known modulators of MITF transcriptional activity (Howlin et al, 2015; Qi et al, 2013), which were significantly upregulated by rTFE3-L and rTFE3-L-M106A, but no rTFE3-S (Fig. 5F). These results were confirmed by immunoblot analysis

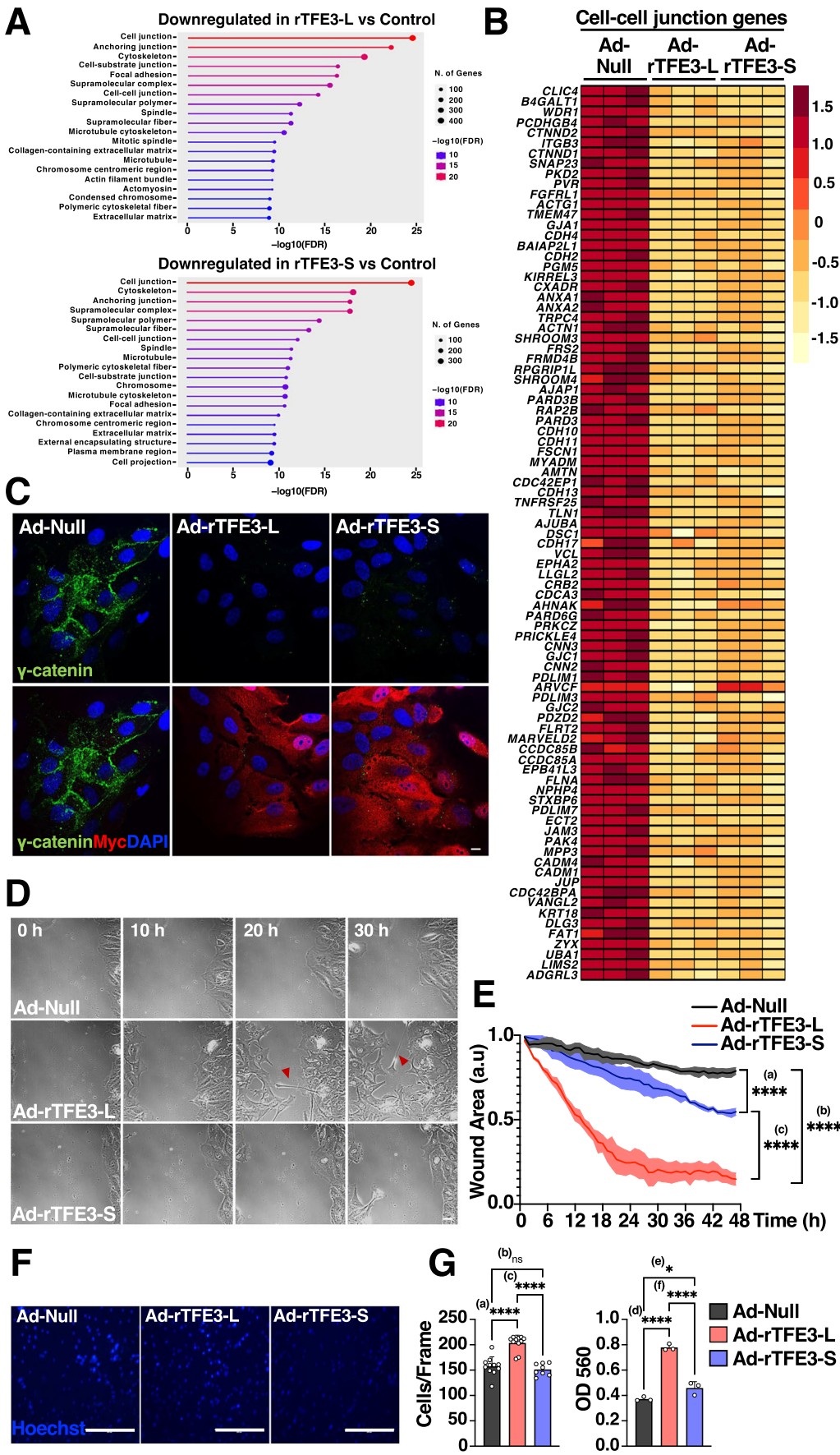

**Figure 4. Cell proliferation, cell–cell contact, and motility are affected upon overexpression of TFE3 isoforms.**

(A) KEGG pathway enrichment analysis of differentially expressed genes in ARPE19 cells infected with either control adenovirus or adenovirus expressing recombinant TFE3-L-Myc or TFE3-S-Myc for 30 h. Circle size represents the number of genes, while the color scale indicates the -log10 of the false discovery rate (FDR). (B) Heatmap of differentially expressed cell–cell junction genes in ARPE19 cells infected with the indicated adenovirus. (C) Immunofluorescence confocal microscopy of ARPE19 cells infected with adenovirus expressing recombinant TFE3-L-Myc, TFE3-S-Myc or Control (Null) for 30 h, showing the cellular distribution of γ-Catenin (green) and recombinants TFE3-L-Myc or TFE3-S-Myc (red). Scale bars: 10 μm. (D) Scratch wound healing assay of ARPE19 cells infected with either adenovirus expressing recombinant TFE3-L-Myc or TFE3-S-Myc for 60 h showing representative images from a timelapse of 48 h. Arrowheads indicate axon-like protrusions. Scale bar: 20 μm. (E) Quantification of scratch assay imaging by percentage wound area, measured every 10 min from phase contrast microscopy of ARPE19 cells infected with adenovirus control or adenovirus expressing either TFE3-L-Myc or TFE3-S-Myc for 60 h as shown in (D). Data are presented as mean ± SD of three independent experiments. $^{****(a)}P < 0.0001$; $^{****(b)}P < 0.0001$; $^{****(c)}P < 0.0001$ (one-way ANOVA followed by Tukey's multiple comparison post-test). (F) Representative images of Hoechst-stained membranes from Boyden chamber invasion assay after removal of non-invading ARPE19 cells infected with adenovirus expressing recombinant TFE3-L-Myc, TFE3-S-Myc or Control for 30 h. Scale bars: 200 μm. (G) Quantification of (F) by nuclear count per frame (540 × 400 μm) and colorimetric measurement of migrating cells stained with crystal violet. Data are presented as mean ± SD of three independent experiments. $^{****(a)}P < 0.0001$; (ns) not significant $^{(b)}P = 0.6721$; $^{****(c)}P < 0.0001$; $^{****(d)}P < 0.0001$; $^{*(e)}P = 0.006$; $^{****(f)}P < 0.0001$ (one-way ANOVA followed by Dunnett's multiple comparison post-test). Source data are available online for this figure.

(Fig. 5G). While all the isoforms induced upregulation of LAMP1 and LC3, the expression of CITED1 and two CEBPA isoforms (p42 and p30) was only increased by rTFE3-L and rTFE3-L-M106A (Fig. 5G,H). This selectivity was also confirmed in other cell types, including HeLa and U2OS cells, where TFE3-L was again more efficient than TFE3-S inducing expression of certain target genes, including *CHI3L1* and *CITED1* (Appendix Fig. S9A,B).

A similar trend was observed for several MITF targets, including genes implicated in fate determination (*TBX6* and *HEY2*), cell transformation (*ETV4* and *BRCA1*), and melanogenesis (*PMEL*, *TYR* and *GPR143*) (Fig. EV5A). It is important to note that the reduced ability of rTFE3-S to upregulate specific targets does not seem to be due to its reduced expression, as the mRNA levels of all the isoforms were comparable (Fig. EV5A). Likewise, it does not seem to reflect decreased transcriptional capability or activation, since rTFE3-S was more efficient than rTFE3-L or rTFE3-L-M106 A in promoting expression of other target genes, such as *SLC45A4* and *ACE2*, and all isoforms caused efficient repression of *ROS* (Fig. EV5A). Differential expression of proteins implicated in terminal melanogenesis was further corroborated by immunoblot and immunofluorescence analysis (Fig. EV5B–D).

Finally, we observed that rTFE3-L was more efficient than rTFE3-S in increasing the expression of several negative regulators of the canonical Wnt pathway, including *KREMEN2*, *ZNRF3*, *AXIN2*, *TLE3*, *NKD1*, *DKK2*, and *SCRT1* (Fig. EV5E), which may explain the slightly stronger effect of rTFE3-L overexpression on proliferation inhibition (Fig. EV4C). Altogether our results suggest that rTFE3-L may have unique properties promoting cellular migration and differentiation and that it may do so, at least in part, by modulating MITF target selectivity.

## TFE3-L promotes cell migration and invasion of tuberous sclerosis complex mutant cells

To further investigate the potential contribution of endogenous TFE3-L to migration and invasion, we analyzed TSC2 knockout cells. Depletion of TSC2, one of the two proteins commonly mutated in Tuberous Sclerosis Complex (TSC), prevents mTORC1-dependent phosphorylation of TFE3, causing its persistent accumulation in the nucleus and potentially preventing the degron-mediated degradation

of TFE3-L (Alesi et al, 2021; Lam et al, 2018). Analysis of a previously described TSC2 knockout HeLa clone (Alesi et al, 2024) confirmed TFE3 activation, as well as the presence of the two TFE3 isoforms (Fig. 6A). In agreement with previous studies reporting a link between TSC2 inactivation and increased metastasis in some instances of PEComas and renal carcinoma (Gupta et al, 2024; Meredith et al, 2023), we found that depletion of TSC2 in HeLa cells significantly increased their motility (Fig. 6B). Notably, targeted CRISPR knockout of TFE3-L, reduced the motility of TSC2-KO cells to levels comparable to HeLa WT (Fig. 6A,B). TSC2-KO cells also presented alterations in the organization of cell–cell adhesion structures. As shown in Fig. 6C, the adherent junctions labeled by N-cadherin displayed a disorganized and radial distribution that resemble those described in transformed cells (Ayollo et al, 2009), a phenotype that was partially rescued by depletion of TFE3-L. Furthermore, the increased invasive properties of TSC2-KO cells were significantly reduced by depletion of TFE3-L (Fig. 6D,E). In summary, our results suggest a potential contribution of TFE3-L to cancer progression.

## Discussion

In this study, we characterized the regulation and function of two TFE3 isoforms that are primarily generated by the use of alternative transcription initiation sites. The long ~82 kDa isoform (TFE3-L) is produced when the transcription starts at the 5'UTR located right before exon 1. This isoform contains a N-terminal degron and undergoes continuous proteosome-mediated degradation under basal conditions. However, inhibition of S47 phosphorylation following stress causes its stabilization and accumulation. The short ~72 kDa isoform (TFE3-S) is generated by an alternative transcription initiation site located in the intron upstream exon 3 and results in a protein that lacks the N-terminal degron and it is therefore constitutively expressed. With the exception of the first 105 amino-terminal residues, both isoforms share the same protein sequence, including the same RBD and 14-3-3 binding regulatory motifs, as well as the same activation, DNA binding, and homo- hetero-dimerization domains. As such, both TFE3-L and TFE3-S exhibit the same mechanism of regulation, in which mTORC1-dependent phosphorylation at S321 (S216 in TFE3-S) induces interaction with

 

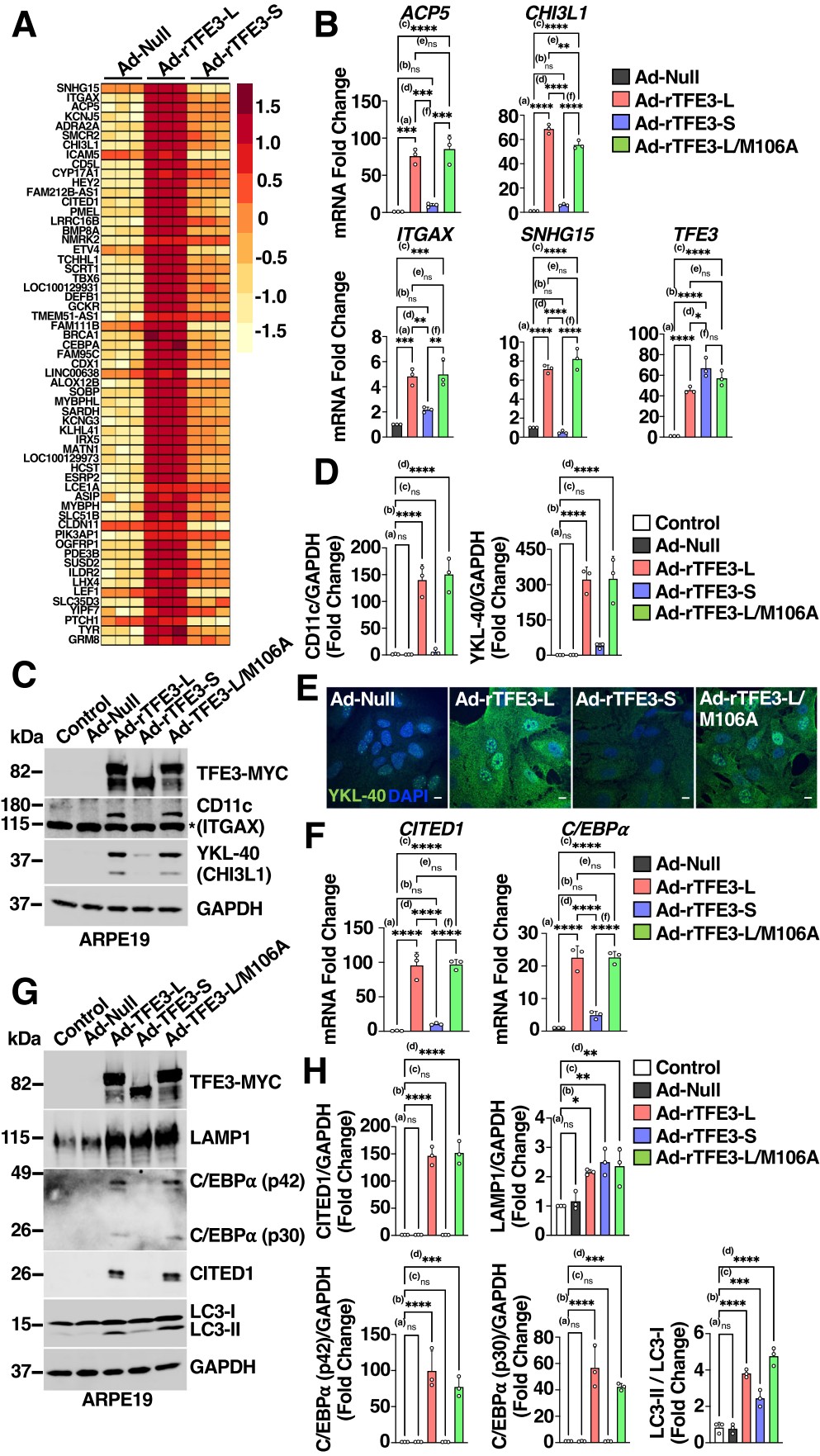

**Figure 5. TFE3-L controls the expression of genes implicated in cellular migration and invasion.**

(A) Heatmap of differentially expressed genes in ARPE19 cells infected with either adenovirus expressing recombinant TFE3-L-Myc or TFE3-S-Myc for 30 h. (B) Relative quantitative RT-PCR analysis of the mRNA expression of the indicated genes in ARPE19 cells infected with either adenovirus expressing recombinant TFE3-L-Myc, TFE3-S-Myc, TFE3-L/M106A-Myc, or Control (Null) for 30 h. Data are presented as mean ± SD of three independent experiments. ACP5 (****[a]$P = 0.0001$; (ns) not significant [b]$P = 0.7598$; ****[c]$P < 0.0001$; ***[d]$P = 0.0003$; (ns) not significant [e]$P = 0.709$; ****[f]$P = 0.0001$), CHI3L1 (****[a]$P < 0.0001$; (ns) not significant [b]$P = 0.1499$; ****[c]$P < 0.0001$; ****[d]$P < 0.0001$; **[e]$P = 0.0011$; ****[f]$P < 0.0001$), ITGAX (***[a]$P = 0.0004$; (ns) not significant [b]$P = 0.1892$; ***[c]$P = 0.0003$; **[d]$P = 0.004$; (ns) not significant [e]$P = 0.9899$; **[f]$P = 0.0028$), SNHG15 (****[a]$P < 0.0001$; (ns) not significant [b]$P = 0.7576$; ****[c]$P < 0.0001$; ****[d]$P < 0.0001$; (ns) not significant [e]$P = 0.2091$; ****[f]$P < 0.0001$), TFE3 (****[a]$P < 0.0001$; ****[b]$P < 0.0001$; ****[c]$P < 0.0001$; *[d]$P = 0.0108$; (ns) not significant [e]$P = 0.1693$; (ns) not significant [f]$P = 0.2653$) (one-way ANOVA followed by Tukey's multiple comparison post-test). (C) Immunoblot analysis of protein lysates from ARPE19 cells infected with the indicated adenovirus for 30 h. Asterisk (*) indicates unspecific band. (D) Quantification of protein levels showing CD11c/GAPDH or YKL-40/GAPDH ratios expressed as fold change as shown in (C). Data are presented as mean ± SD of three independent experiments. (ns) not significant [a]$P > 0.9999$; ****[b]$P < 0.0001$; (ns) not significant [c]$P = 0.9986$; ****[d]$P < 0.0001$ (one-way ANOVA followed by Dunnett's multiple comparison post-test). (E) Immunofluorescence confocal microscopy of ARPE19 cells infected with adenovirus expressing recombinant TFE3-L-Myc, TFE3-S-Myc, TFE3-L/M106A-Myc or Control for 30 h, showing the cellular distribution of YKL-40 (green). Scale bars: 10 μm. (F) Relative quantitative RT-PCR analysis of the mRNA expression of transcription regulator genes (*CITED1* and *C/EBPα*) in ARPE19 cells infected with adenovirus expressing recombinant TFE3-L-Myc, TFE3-S-Myc, TFE3-L/M106A-Myc or Ad-Null for 30 h. Data are presented as mean ± SD of three independent experiments. CITED1 (****[a]$P < 0.0001$; (ns) not significant [b]$P = 0.1888$; ****[c]$P < 0.0001$; ****[d]$P < 0.0001$; (ns) not significant [e]$P > 0.9999$; ****[f]$P < 0.0001$), C/EBPα (****[a]$P < 0.0001$; (ns) not significant [b]$P = 0.1888$; ****[c]$P < 0.0001$; ****[d]$P < 0.0001$; (ns) not significant [e]$P > 0.9999$; ****[f]$P < 0.0001$) (one-way ANOVA followed by Tukey's multiple comparison post-test). (G) Immunoblot analysis of protein lysates from ARPE19 cells infected with adenovirus expressing recombinant TFE3-L-Myc, TFE3-S-Myc, TFE3-L/M106A-Myc, or Ad-Null for 30 h. (H) Quantification of protein levels showing CITED1/GAPDH, LAMP1/GAPDH, C/EBPα-p42/GAPDH, C/EBPα-p30/GAPDH, and LC3-II/LC3-I ratios expressed as fold change as shown in (G). Data are presented as mean ± SD of three independent experiments. CITED1 ((ns) not significant [a]$P > 0.9999$; ****[b]$P < 0.0001$; (ns) not significant [c]$P > 0.9999$; ****[d]$P < 0.0001$), C/EBPα (p42) ((ns) not significant [a]$P > 0.9999$; ****[b]$P < 0.0001$; (ns) not significant [c]$P > 0.9999$; ***[d]$P = 0.0002$), LAMP1 ((ns) not significant [a]$P = 0.956$; *[b]$P = 0.0127$; **[c]$P = 0.0023$; **[d]$P = 0.0044$), C/EBPα (p30) ((ns) not significant [a]$P > 0.9999$; ****[b]$P < 0.0001$; (ns) not significant [c]$P > 0.9999$; ****[d]$P = 0.0001$), LC3 ((ns) not significant [a]$P = 0.8551$; ****[b]$P < 0.0001$; ***[c]$P = 0.0009$; ****[d]$P < 0.0001$) (one-way ANOVA followed by Dunnett's multiple comparison post-test). Source data are available online for this figure.

14-3-3 and consequent cytosolic sequestration. A recent study suggested that the modulation of TFE3-L protein stability may constitute its main method of regulation (Nardone et al, 2023). However, we show that even when TFE3-L is expressed at high levels, the protein is still retained in the cytosol as long as S321 remains phosphorylated.

Remarkably, both isoforms displayed comparable efficiency in promoting upregulation of lysosomal and autophagic genes in response to activation and, at least under overexpression conditions, both isoforms precluded cell-cycle progression and reduced cellular proliferation. These observations raise an obvious question, if TFE3-S, which is constitutively expressed at high levels, has the ability to target the same cellular pathways as TFE3-L, what is the reason for the continuous synthesis and degradation of TFE3-L? One possibility is that despite the energetic disadvantage of constantly eliminating TFE3-L, this may allow cells to quickly respond to acute stress conditions, as high TFE3-L mRNA levels would be always present. However, this does not seem to be the case. When we incubated ARPE19 cells in EBSS medium, we observed a rapid and robust TFE3-S activation, while it took several hours to start observing a significant TFE3-L accumulation.

Another possibility is that TFE3-L present specific properties that will only be required after prolonged stress conditions or following sustained mTORC1 inhibition. In agreement with this idea, we observed that TFE3-L was considerably more efficient than TFE3-S in promoting cell migration and invasion. This might have important consequences in cancer progression. Tumors often encounter nutrient limitations that drive them to adopt migratory phenotypes, and starvation-mediated autophagy has been linked to metastasis in hepatocellular carcinoma, bladder and ovary cancer (Li et al, 2013; Lin et al, 2023; Tong et al, 2019). Consistently, we found that specific depletion of TFE3-L in a cellular model for tuberous sclerosis complex, a rare disease characterized by the growth of tumors in multiple organs, significantly reduced cells migration and invasion. However, a limitation of our work is that

we only use cellular models. Future studies will need to address the potential contribution of two TFE3 isoforms to cancer progression and metastasis in vivo.

How does TFE3-L acquire its unique properties? An unexpected observation was the differential target selectivity displayed by both isoforms regarding a specific subset of genes, including some implicated in metastasis and EMT. Given that the two isoforms share the same activation and DNA binding motifs, it is likely that the extra N-terminal 105 residues present in TFE3-L confers specificity by binding particular transcriptional regulators. One of the genes that was specifically upregulated by TFE3-L was CITED1, a critical regulator of MITF transcriptional activity. MITF functions as a rheostat, meaning that its level of activity determines its role in different cellular processes. In melanoma, high MITF activity has been associated with proliferative, non-invasive phenotypes, while melanoma cells with low MITF activity show increased metastatic and invasive capacity (Carreira et al, 2006; Goodall et al, 2008; Hoek et al, 2008). It was reported that CITED1 functions as a phenotypic switch to reduce expression of MITF proliferative targets relative to invasive targets (Howlin et al, 2015). Therefore, by regulating CITED1 levels, TFE3-L might indirectly affect expression of specific MITF targets. Increased CITED1 expression also enhances tumorigenicity in childhood kidney cancers, such as Wilms tumor (Murphy et al, 2014), and increases metastasis in breast cancer (Chen et al, 2024). In certain cellular contexts, CITED1 represses canonical Wnt, a pathway that induces cell proliferation while inhibiting cell migration (Plisov et al, 2005). CEBPA, another of the genes preferentially induced by TFE3-L, has also been shown to have an antagonist effect on Wnt signaling (Zhang et al, 2021). This is consistent with our results showing that TFE3-L favors expression of several negative regulators of the canonical Wnt pathway, including DKK2, a protein that antagonizes canonical Wnt signaling while promoting the migration-inducer non-canonical Wnt pathway (Park et al, 2022). Our data are also in agreement with a recent report showing

 

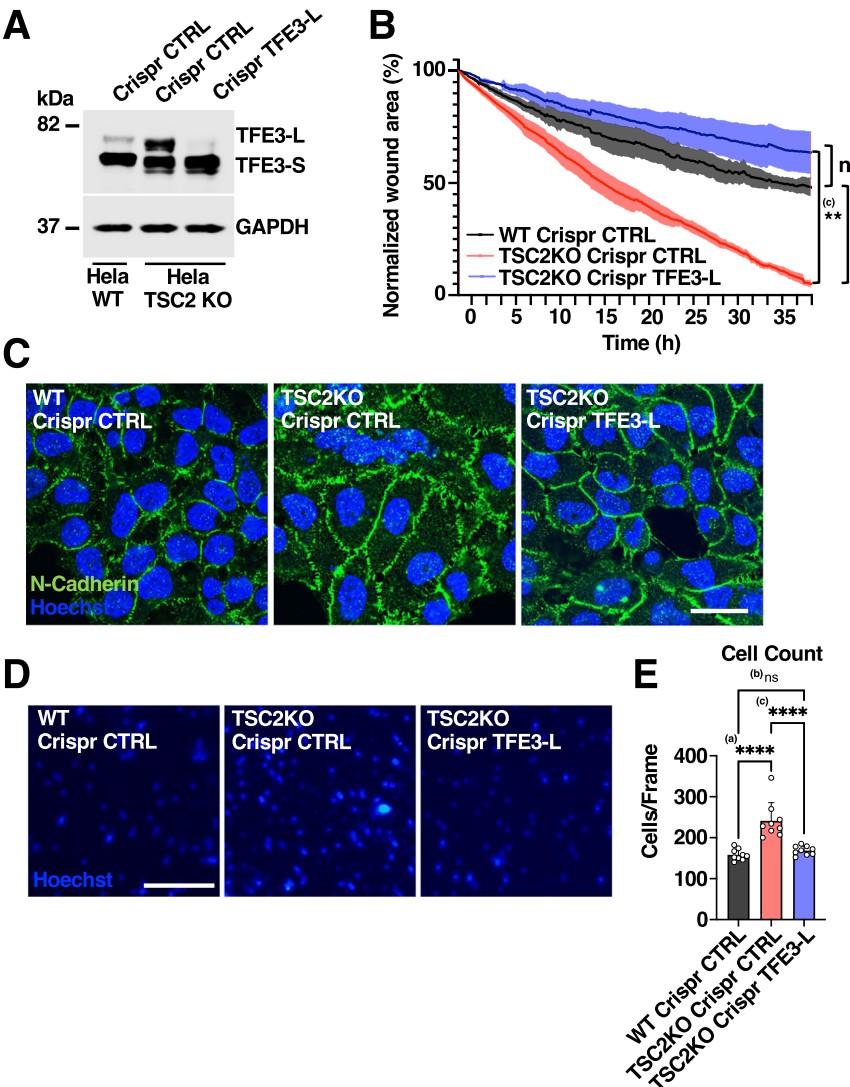

**Figure 6. TFE3-L knockout reduces motility and invasiveness of TSC2 mutant cells.**

(A) Immunoblot analysis of protein lysates from HeLa WT or HeLa TSC2 knockout cells expressing either non-targeting (Control) or TFE3-L targeting gRNA. (B) Quantification of scratch assay imaging by percentage wound area, measured every 10 min from phase contrast microscopy of WT Crispr control, TSC2KO Crispr control, and TSC2KO Crispr TFE3-L HeLa cells. Data are presented as mean ± SD of three independent experiments. *(a)P = 0.022; (ns) not significant (b)P = 0.3184; **(c)P = 0.0042 (one-way ANOVA followed by Tukey's multiple comparison post-test). (C) Immunofluorescence confocal microscopy of HeLa control or TSC2KO CRISPR cells showing the cellular distribution of N-Cadherin (green). Scale bars: 25 µm. (D) Representative images of Hoechst-stained membranes from Boyden chamber invasion assay after removal of non-invading WT Crispr control, TSC2KO Crispr control, and TSC2KO Crispr TFE3-L cells. Scale bars: 200 µm. (E) Quantification of (D) by nuclear count per frame (540 × 400 µm). Data are presented as mean ± SD of three independent experiments. ****(a)P < 0.0001; (ns) not significant (b)P = 0.7101; ****(c)P < 0.0001 (one-way ANOVA followed by Tukey's multiple comparison post-test). For each experimental replicate, three images of the membrane were counted. Source data are available online for this figure.

TFE3-dependent inhibition of canonical Wnt signaling in FLCN-deficient cells (Kennedy et al, 2019). Therefore, we suggest that the overall effect of TFE3-L on migration likely result from a combination of different factors, including alterations in Wnt signaling, expression of direct TFE3-L targets, and modulation of MITF selectivity.

In summary, our data indicates that TFE3-L and TFE3-S exhibit partial redundancy. Both isoforms increased expression of lysosomal and autophagy genes, reduced proliferation, and

induced reorganization of cell–cell interactions. However, TFE3-L was more efficient than TFE3-S promoting migratory and invasive behaviors. Conversely, the early response to a variety of stress conditions, including nutrient deprivation, oxidative stress, and lysosomal damage, seemed to be largely mediated by TFE3-S, suggesting a major contribution of this isoform in restoration of energy homeostasis and cell survival. Our work provides new insight into the intricacies of TFE3 function and regulation.

# Methods

**Reagents and tools table**

| Reagent or resource | Source | Identifier |
| --- | --- | --- |
| **Antibodies** | | |
| TFE3 Antibody, Mouse monoclonal | Sigma-Aldrich | SAB4200824-100U, RRID:AB_3683106 |
| Phospho-4E-BP1 (Ser65) Antibody | Cell Signaling Technology | Cat# 9451, RRID:AB_330947 |
| 4E-BP1 (53H11) Rabbit mAb | Cell Signaling Technology | Cat# 9644, RRID:AB_2097841 |
| GAPDH Monoclonal Antibody (6C5) | Thermo Fisher Scientific | Cat# AM4300, RRID:AB_2536381 |
| FLCN (D14G9) Rabbit mAb | Cell Signaling Technology | Cat# 3697, RRID:AB_2231646 |
| Anti-phospho S321 TFE3 | Martina et al, 2016 | N/A |
| TFE3 Antibody | Cell Signaling Technology | Cat# 14779, RRID:AB_2687582 |
| HA-Tag (C29F4) Rabbit mAb | Cell Signaling Technology | Cat# 3724, RRID:AB_1549585 |
| Histone H3 (D1H2) XP Rabbit mAb | Cell Signaling Technology | Cat# 4499, RRID:AB_10544537 |
| Myc-Tag Antibody | Cell Signaling Technology | Cat# 2272, RRID:AB_10692100 |
| Myc-Tag (9B11) Mouse mAb | Cell Signaling Technology | Cat# 2276, RRID:AB_331783 |
| TFE3 antibody (N-Term) | Antibodies Online | Cat# ABIN1538859, RRID:AB_3683105 |
| Goat anti-Rabbit IgG (Heavy Chain), Superclonal Recombinant Secondary Antibody, Alexa Fluor™ 488 | Thermo Fisher Scientific | Cat# A27034, RRID:AB_2536097 |
| Goat anti-Rabbit IgG (Heavy Chain), Superclonal Recombinant Secondary Antibody, Alexa Fluor™ 555 | Thermo Fisher Scientific | Cat# A27039, RRID:AB_2536100 |
| Anti-rabbit IgG, HRP-linked Antibody | Cell Signaling Technology | Cat# 7074, RRID:AB_2099233 |
| Anti-mouse IgG, HRP-linked Antibody | Cell Signaling Technology | Cat# 7076, RRID:AB_330924 |
| γ-Catenin (D9M1Q) Rabbit mAb | Cell Signaling Technology | Cat# 75550, RRID:AB_2799872 |
| β-Catenin (D10A8) XP Rabbit mAb | Cell Signaling Technology | Cat# 8480, RRID:AB_11127855 |
| p21 Waf1/Cip1 (12D1) Rabbit mAb | Cell Signaling Technology | Cat# 2947, RRID:AB_823586 |
| Rb (4H1) Mouse mAb | Cell Signaling Technology | Cat# 9309, RRID:AB_823629 |
| Phospho-Rb (Ser807/811) (D20B12) XP Rabbit mAb | Cell Signaling Technology | Cat# 8516, RRID:AB_11178658 |
| CD11c (D3V1E) XP Rabbit mAb | Cell Signaling Technology | Cat# 45581, RRID:AB_2799286 |
| YKL-40 (E2L1M) Rabbit mAb | Cell Signaling Technology | Cat# 47066, RRID:AB_2799320 |

| Reagent or resource | Source | Identifier |
| --- | --- | --- |
| LAMP-1 (human) | DSHB | Cat# H4A3, RRID:AB_2296838 |
| CITED1 Polyclonal antibody | Proteintech | Cat# 26999-1-AP, RRID:AB_2296838 |
| C/EBPα (D56F10) XP Rabbit mAb | Cell Signaling Technology | Cat# 8178, RRID:AB_11178517 |
| LC3B Antibody | Cell Signaling Technology | Cat# 2775, RRID:AB_915950 |
| MLANA/MART-1 Antibody | Cell Signaling Technology | Cat# 34511, RRID:AB_2799055 |
| PMEL Monoclonal Antibody (HMB45) | Thermo Fisher Scientific | Cat# MA1-34759, RRID:AB_1955861 |
| Tyrosinase Monoclonal Antibody (T311) | Thermo Fisher Scientific | Cat# MA5-14177, RRID:AB_10980000 |
| N-Cadherin (D4R1H) XP Rabbit mAb | Thermo Fisher Scientific | Cat# 13116, RRID:AB_2687616 |
| ATG7 Antibody | Cell Signaling Technology | Cat# 2631 RRID:AB_2227783 |
| **Bacterial and virus strains** | | |
| pMXs-Puro | Cell Biolabs | RTV-012 |
| Adenovirus | Welgen, Inc | N/A |
| **Chemicals, peptides, and recombinant proteins** | | |
| Torin 1 | Tocris | Cat# 4247 |
| MLN4924 (Pevonedistat) | Selleckchem | Cat# S7109 |
| SYBR GreenER™ | Thermo Fisher Scientific | Cat# 11760100 |
| MG132 | Cell Signaling Technology | Cat# 2194 |
| NaAsO2 | Santa Cruz Biotechnology | Cat# sc-250986 |
| CCCP | Cayman Chemical | Cat# 25458 |
| LLOMe | Cayman Chemical | Cat# 16008 |
| Chloroquine | Sigma-Aldrich | Cat# C6628 |
| LPS | InvivoGen | Cat# tlrl-3pelps |
| Human fibronectin | Millipore Sigma | Cat# FC010 |
| Alt-R™ S.p. Cas9 Nuclease V3 | IDT | Cat# 1081059 |
| Opti-MEM™ I Reduced Serum Medium | Thermo Fisher Scientific | Cat# 31985062 |
| Lipofectamine™ RNAiMAX Transfection Reagent | Thermo Fisher Scientific | Cat# 13778075 |
| Hoechst 33342 Solution (20 mM) | Thermo Fisher Scientific | Cat# 62249 |
| Pierce™ IP Lysis Buffer | Thermo Fisher Scientific | Cat# 87788 |

| Reagent or resource | Source | Identifier |
|---|---|---|
| EBSS, calcium, magnesium, phenol red | Thermo Fisher Scientific | Cat# 24010043 |
| cOmplete™, Mini, EDTA-Free protease inhibitor Cocktail | Millipore Sigma | Cat# 11836170001 |
| PhosSTOP phosphatase Inhibitor Cocktail | Millipore Sigma | Cat# 4906845001 |
| NuPAGE™ LDS Sample Buffer (4X) | Thermo Fisher Scientific | Cat# NP0007 |
| NuPAGE™ Sample Reducing Agent (10X) | Thermo Fisher Scientific | Cat# NP0009 |
| Nitocellulose/Filter Paper Sandwich, 0.2 µm, 8.3 ×7.3 cm | Thermo Fisher Scientific | Cat# LC2000 |
| DMEM/F-12, GlutaMAX™ supplement | Thermo Fisher Scientific | Cat# 10565018 |
| DMEM, high glucose, GlutaMAX™ Supplement, pyruvate | Thermo Fisher Scientific | Cat# 1056944 |
| Glutathione-Sepharose beads | GE Healthcare | Cat# 17513201 |
| **Critical commercial assays** | | |
| PureLink™ RNA Mini Kit | Thermo Fisher Scientific | Cat# 12183018 A |
| SuperScript™ III First-Strand Synthesis SuperMix for qRT-PCR | Thermo Fisher Scientific | Cat# 11752050 |
| QuikChange Lightning Site-Directed Mutagenesis Kit | Agilent Technologies | Cat# 210518-5 |
| Click-IT™ Plus EdU Alexa Fluor™ 488 Flow Cytometry Assay Kit | Thermo Fisher Scientific | Cat# C10632 |
| Cell line Nucleofactor™ Kit V | Lonza | Cat# VCA-1003 |
| P3 Primary Cell 4D-Nucleofactor™ X Kit S | Lonza | Cat# V4XP-3032 |
| **Experimental models: cell lines** | | |
| ARPE19 cells | American Type Culture Collection | Cat# CRL-2302 |
| HEK293T | American Type Culture Collection | Cat# CRL-3216 |
| RAW264.7 | American Type Culture Collection | Cat# TIB-71 |
| U2OS | American Type Culture Collection | Cat# HTB-96 |
| HeLa | American Type Culture Collection | Cat# CRM-CCL-2 |
| Hela TSC2-KO cells | Alesi et al, 2024 | N/A |

| Reagent or resource | Source | Identifier |
|---|---|---|
| **Oligonucleotides** | | |
| See Table EV1 for a list of oligonucleotides | N/A | N/A |
| **Software and algorithms** | | |
| FlowJo 10.8.1 | Becton, Dickinson & Company | https://www.flowjo.com/ |
| GraphPad Prism 9.4.1 | GraphPad | https://www.graphpad.com/scientific-software/prism/ |
| Fiji | Schneider et al, 2012 | https://imagej.net/Fiji |
| SoftMax Pro Software | Molecular Devices | https://www.moleculardevices.com/ |
| ShinyGO 0.82 | Ge et al, 2020 | http://bioinformatics.sdstate.edu/go/ |
| QuantStudio 12 K Software v1.5 | Thermo Fisher Scientific | https://www.thermofisher.com/ |

## Cell lines and cell culture

ARPE19 cells (CRL-2302, American Type Culture Collection) were cultured in a 1:1 mixture of DMEM and Ham's F12 media with GlutaMAX™ (Thermo Fisher Scientific, Cat# 10565018). HEK293T (CRL-3216, American Type Culture Collection), RAW264.7 (TIB-71, American Type Culture Collection), U2OS (HTB-96, American Type Culture Collection), HeLa (CRM-CCL-2, American Type Culture Collection) and Hela TSC2-KO cells (previously described in (Alesi et al, 2024)) were grown in DMEM GlutaMAX™ (Thermo Fisher Scientific, Cat# 10569010) at 37 °C in 5% $CO_2$. Cell culture media was supplemented with 10% (v/v) FBS (Thermo Fisher Scientific, Cat# 10438034), 1% (v/v) penicillin-streptomycin (Thermo Fisher Scientific, Cat# 15140122). All cell lines were tested for Mycoplasma contamination.

## Recombinant DNA plasmid and adenovirus production

Recombinant TFE3-MYC expression vector has been previously described (Martina et al, 2014). TFE3-S-Myc expression vector was generated by cloning the amino acid sequence corresponding to Met106 to Ser575 of human TFE3. The cDNA fragment obtained by PCR amplification from pCMV-3TAG-4A-TFE3 (Martina et al, 2014) was cloned in-frame into BamHI and SalI sites of pCMV-3TAG-4A (Agilent Technologies) with a triple Myc tag fused to the carboxy-termini of TFE3-S using In-fusion HD EcoDry system (Takara Bio USA, Inc.). Amino acid substitutions in TFE3 were made using the QuikChange Lightning site-directed mutagenesis kit (Agilent Technologies) according to the manufacturer's instructions. Constructs were confirmed by DNA Sanger sequencing. Adenovirus expressing rTFE3-L-Myc, rTFE3-S-Myc and rTFE3-L/M106A-Myc were produced, amplified and purified by Welgen, Inc. Adenovirus expressing TFE3-WT-Myc has been previously described (Martina et al, 2014).

## Recombinant DNA transfection and adenovirus infection

For transient expression, ARPE19 cells were nucleofected using Cell Line Nucleofector® Kit V in a Nucleofector 2B system (Lonza) following manufacturer's recommendations. Cells were analyzed 24 h post-nucleofection. For infection experiments, ARPE19 cells were infected with adenoviruses according to the manufacturer's recommendations. Analyses were performed 30-48 h post-infection.

## Generation of TSC2 KO Crispr TFE3-L-Knock down (KD) cell line

CRISPR-Cas9 technology was used to obtain the HeLa TSC2 KO Crispr TFE3-L-KD cell line. Guide RNAs (crRNA) were designed using the design tool from Integrated DNA Technologies, Inc (IDT). A specific TFE3-L crRNA targeting the end of exon 1 or a negative control (CTRL) were mixed at 1:1 ratio with tracrRNA and heated at 95 °C for 5 min to obtain the crRNA-tracrRNA complex. Then, Cas9 enzyme and Duplex buffer (IDT, Inc) were added and incubated at room temperature for 10 min to allow the formation of the crRNA-tracrRNA-Cas9 complex (RNP). Amaxa P3 primary cell 4D-Nucleofactor X kit (Lonza, Cat# V4XP-3032) and 4D-Nucleofector® System were used to electroporate the RNPs into TSC2 KO cells ($0.5 \times 10^6$ cells/cuvette) using program CN-114. Electroporated cells were seeded in six-well plates and expanded for 72 h before analysis.

## RNA interference transfection (siRNA)

ARPE19 cells grown in a six-well plate were transfected with Lipofectamine™ RNAiMAX Transfection Reagent (Thermo Fisher Scientific) and 100 nM of ON-TARGETplus non-targeting pool siRNA duplexes or ON-TARGETplus smart pool siRNA duplexes targeted against human FLCN and ATG7 (Horizon Discovery). Treated cells were processed for analysis 72 h after transfection.

## Subcellular fractionation

Cells were washed with ice-cold PBS, scraped gently with 1 mL of PBS, and centrifuged at 2000 x rpm for 2 min. Cell pellets were resuspended and incubated with 200 μL of buffer A (20 mM Tris-HCl pH 7.6, 0.1 mM EDTA, 2 mM $MgCl_2$ containing protease and phosphatase inhibitors cocktail) for 2 min at room temperature. Then samples were incubated for 10 min on ice. Subsequently, 1% Igepal CA-630 (Sigma-Aldrich Cat# I3021) was added, samples were passed three times through a 20G-needle syringe and centrifuged at 500 x g for 3 min at 4 °C. The resulting supernatants (cytoplasmic fractions) were collected, and the corresponding cells pellets were washed 3 times with buffer A + 1% Igepal CA-630 by centrifugation at $500 \times g$ for 3 min at 4 °C. Then, pellets were treated with 100 μL of buffer B (20 mM HEPES pH 7.9, 400 mM NaCl, 2.5% (V/V) glycerol, 1 mM EDTA and 0.5 mM DTT containing protease and phosphatase inhibitors cocktail). The resuspended pellets were snap-frozen in liquid nitrogen for 30 s and then defrosted at 37 °C. This step was repeated 4 times. The samples were incubated for 20 min on ice and centrifuged at 20,000 $\times g$ for 20 min. The corresponding supernatants represent the nuclear fraction.

## Electrophoresis and immunoblotting and GST-pull-down

For the preparation of total lysates, ARPE19, HEK293T, U2OS, RAW264.7, HeLa WT and HeLa TSC2-KO cells were washed with ice-cold PBS and then lysed with Lysis Buffer (Thermo Fisher Scientific, Cat# 87788) containing protease and phosphatase inhibitors cocktail (Thermo Fisher Scientific, Cat# 78440). The lysates were incubated for 20 min on ice and then centrifuged at 14,000 $\times g$ for 30 min at 4 °C. For GST-pull-down, soluble fractions were incubated with 25 μl of Glutathione-Sepharose (GE Healthcare Cat# 17513201) beads for 2 h at 4 °C. Proteins were quantified by the bicinchoninic acid method (Thermo Fisher Scientific, Cat# 23225). Forty micrograms of proteins were boiled for 10 min in the presence of NuPAGE™ LDS Sample Buffer (Thermo Fisher Scientific, Cat# NP0007) and NuPAGE™ Sample Reducing Agent (Thermo Fisher Scientific, Cat# NP0009). Proteins were subjected to SDS-PAGE and transferred to nitrocellulose membranes (Thermo Fisher Scientific, Cat# LC2000). Membranes were then blocked with either nonfat dry milk 5% (Bio-Rad, Cat#1706404) or Bovine Serum Albumin 5% (Sigma-Aldrich, Cat# A3294) in Tris-buffered saline 0.1% Tween 20 detergent (TTBS) for 1 h at RT. Blocked membranes were incubated with primary antibody overnight at 4 °C. Then, membranes were washed four times for 5 min with TTBS and incubated with HRP-conjugated secondary antibody (1:5000) for 1 h at room temperature with shaking. Detection was performed with Femtogram HRP substrate Radiance Plus (Azure Biosystems, Cat#AC2103) using the C300 gel imaging system for chemiluminescence (Azure Biosystems). Fiji software was used for quantification by densitometry.

## Immunofluorescence confocal microscopy

Cells were seeded on coverslips in 24-well culture plates. After treatments, cells were washed with PBS and fixed with 4% paraformaldehyde for 20 min at RT. Then, cells were washed with PBS, permeabilized with 0.2% Triton X-100 for 10 min at RT, and incubated with the indicated primary antibodies in PBS containing 10% Fetal Bovine Serum and 0.1% (wt/v) saponin for 1 h at RT or overnight at 4 °C, followed by incubation with the corresponding Alexa Fluor 568-conjugated and Alexa Fluor 488-conjugated secondary antibodies. After staining, the coverslips were mounted onto glass slides with Dapi-Fluoromount-G (Electron Microscopy Sciences, 17984-24). Images were acquired on a Zeiss LSM 780 confocal system equipped with filter sets for Alexa 488, 568 and DAPI, 561 nm, 488 nm and 405 nm laser excitations, respectively, 63×/1.4 Oil Zeiss Plan-Apochromat 63× NA 1.4 oil immersion objective, and ZEN black imaging software (Carl Zeiss). Confocal images taken with the same acquisition parameters were processed with ImageJ software (NIH). Photoshop 2025 software was used to produce the figures.

## RNA isolation, PCR amplification and relative quantitative Real-Time PCR

Total RNA was extracted from cells with the PureLink™ RNA Mini Kit (Thermo Fisher Scientific). Reverse transcription reactions were carried out using SuperScript™ III First-Strand Synthesis SuperMix for qRT-PCR (Thermo Fisher Scientific) following the manufacturer's instructions. Quantitative real-time PCR reactions were set

 

up in triplicate with 50 ng cDNA per reaction and 200 nM gene specific primers mix (QuantiTect primer Assays, Qiagen) along with SYBR GreenER™ (Thermo Fisher Scientific). Reactions were run and analyzed using a QuantStudio™ 12 K Flex real-time PCR system (Applied Biosystems, Life Technologies). The values were expressed as a fold change relative to RNA from cells infected with control adenovirus (Ad. Null) or mock transfected cells, normalized against GAPDH using the ΔΔCT methods. For each gene, 3 technical replicates and 3 biological samples were analyzed. Primers used in this study are listed in Table EV1.

For the PCR amplification of different TFE3 fragments, 500 ng of either cDNA retrotranscribed from RNA or genomic DNA (gDNA) from ARPE19 cells was added to a master-mix of Phusion™ High-Fidelity DNA Polymerase (1 unit, NEB #M0530), dNTPs (200 μM) and Phusion buffer containing the specific forward and reverse primers (0.5 μM each) in a final reaction volume of 50 μl. Primers used are listed in Table EV1. PCR was performed using a SimpliAmp thermal cycler (Fisher Scientific) under the following conditions: initial denaturation (98 °C for 30 s), followed by 35 cycles of denaturation (98 °C for 10 s), annealing (65 °C for 30 s) and extension (72 °C for 30 s) and a final extension at 72 °C for 5 min. Ten microliters of the PCR products were run on 1.5% SeaKem® LE agarose gel (Lonza) in 1× TAE buffer (40 mM Tris-acetate, 1 mM EDTA, pH 8.3). Gel images were acquired using C300 gel imaging system with the DNA detection mode (Azure Biosystems).

## Cloning and sequencing of PCR products

The PCR products amplified from TFE3 cDNA and run in agarose gel were excised, extracted, purified and subcloned into pJET1.2/blunt cloning vector using the CloneJET PCR Cloning Kit (Thermo Fisher Scientific) according to the manufacturer's recommendations. Followed by bacterial transformation, 5 isolated colonies were picked, and the bacteria were grown overnight. Plasmid DNA was extracted using the Wizard® Plus SV Minipreps DNA Purification System (Promega Corporation) and Sanger sequenced (ACGT, Inc) using the corresponding pJET1.2 Forward and Reverse sequencing primer (Thermo Fisher Scientific). Sequences were analyzed using SnapGene software (GSL Biotech, LLC).

## RNA-seq

ARPE19 cells were infected with adenovirus Null, adenovirus rTFE3-L-Myc or adenovirus rTFE3-S-Myc. Thirty hours later, cells were washed two times with PBS, trypsinized, and transferred to a 15 mL conical tube and centrifuged at 800 × g for 5 min. Cell pellets were resuspended with ice-cold PBS and centrifuged again at 800 × g for 5 min. Washed cell pellets were snap-frozen on dry ice and then processed for RNA-seq assay by Active Motif, Inc. For each sample, 0.5 ng of total RNA was used in Illumina's TruSeq Stranded mRNA Library kit (Illumina, Inc). Libraries were sequenced on Illumina NextSeq 500 as paired-end 42-nt reads. Sequence reads were analyzed with the STAR alignment – DESeq2 software pipeline.

## Scratch wound healing assay

For cell migration measurements, cells were seeded in six-well glass bottom plates coated with 10 μg/ml human fibronectin (Millipore, #FC010) and allowed to reach confluence. Prior to imaging, cells were scratched with a sterile micropipette tip and washed twice to remove debris. Phase contrast time-lapse imaging at 10 min intervals for a total of 48 h was performed on a Nikon Ti-E inverted microscope system equipped with the Perfect Focus System, using a Plan Apo 20 × 0.75NA Phase objective lens on a CoolSnap Myo cooled CCD camera (Photometrics), automated X-Y stage with linear encoders (Applied Scientific Instruments, Eugene OR). Cells were maintained at 37 °C and 5% CO$_2$ with a Tokai Hit STXF TIZWX Incubation System. Illumination, image acquisition, and microscope function were controlled by NIS-Elements Software (Nikon). Time-lapse images were analyzed on ImageJ (NIH) with the Wound Healing size tool plugin where measurement of total wound area, area percentage, and standard deviations were collected at each time point. For ARPE19 cells, adenovirus infection was performed 12 h prior to imaging.

## Boyden chamber invasion assay

To measure the invasiveness of CRISPR HeLa and ARPE19 cells, equal density of cells was seeded in serum free DMEM media into an insert chamber of QCM ECMatrix Cell Invasion Assay (Millipore, #ECM550) with 8μm pore. Inserts were placed in a culture plate with serum containing media and allowed to migrate for 24 h or 48 h, respectively. Prior to incubation, ARPE19 cells were infected with adenoviruses expressing either recombinant TFE3-L-Myc or TFE3-S-Myc or Control while in suspension in serum free media and immediately seeded into the inserts. After incubation, cells were fixed in 4% paraformaldehyde and stained with Hoechst 33342 (Thermo Fisher Scientific, #62249). Cells in the top chamber were removed with sterile cotton swabs, and membrane was imaged in multiple locations using an Invitrogen EVOS FL Digital Inverted Fluorescence Microscope at ×20 magnification. Images were identically leveled, and nuclei were counted in ImageJ. Alternatively, colorimetric measurements were collected by removing cells from the top layer and immediately treating the membranes with Millipore Cell Stain solution to fix and stain cells (#90144-200KL). Membranes were dried and subsequently dissolved in 10% acetic acid. Absorbance measurements were then taken of the dissolved membranes at 560 nm using a SpectraMax ID3 Microplate reader (Molecular Devices).

## Flow cytometric quantifications

ARPE19 cells were incubated with Cellstripper™ (Corning, Cat# 25-056-CI) for 15 min at 37 °C. Then, cells were treated with Click-IT™ Plus EdU Alexa Fluor™ 488 Flow Cytometry Assay Kits (Thermo Fisher Scientific, Cat# C10632) according to the manufacturer's instructions. Labeled cells were analyzed in a BD Fortessa cytometer. Quantification of percentage of differentially labeled cells was performed using FlowJo 10.8.1 software.

## Graphical gene-set enrichment and heatmap representation

Gene set enrichment analysis was obtained using the web-based enrichment tool ShinyGO 0.82 (Ge et al, 2020) and heatmap representation were obtained using pheatmap package in R software (The R Project for Statistical Computing).

     

## Quantification and statistical analysis

Data were processed in Excel (Microsoft Corporation) then Prism (GraphPad Software) to generate curve and bar charts and perform statistical analyses. One-way ANOVA with Dunnett's and Tukey's as a post-test. Two-way ANOVA with Tukey's as a post-test. Also, Student's $t$ tests were performed to compare two populations. All data are presented as mean $\pm$ SD. $P < 0.05$ was considered statistically significant (*), $P < 0.01$ very significant (**), $P < 0.001$ extremely significant (***), and $P < 0.0001$ extremely significant (****). $P > 0.05$ was considered not significant (*ns*).

## Data availability

RNA-Seq data have been deposited at GEO and assigned the identifier GSE294493. The source data corresponding to Movies EV1–3 of this paper are collected in the following database record: BioImages accession number S-BIAD2263.

The source data of this paper are collected in the following database record: biostudies:S-SCDT-10_1038-S44319-025-00659-3.

## Peer review information

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

## Acknowledgements

This research was supported by the Intramural Research Program of the National Institutes of Health (NIH). The contributions of the NIH authors are considered Works of the United States Government. The findings and conclusions presented in this paper are those of the authors and do not necessarily reflect the views of the NIH or the U.S. Department of Health and Human Services. Special thanks to Dr. Elizabeth Henske, from Brigham and Women's Hospital, for providing the HeLa TSC2-KO cells. We thank Drs. Pradeep Dagur and Maria-Lopez Ocasio (Flow Cytometry Core, NHLBI) for their assistance performing cell proliferation analysis, members of Dr. Clare Waterman's laboratory (NHLBI), including William Shin and Dr. Ana Pasapera for their assistance in the scratch wound healing and Boyden chamber invasion assays, and Dr. Ji Eun Lee (NIDDK) for her assistance in the bioinformatic analysis. We also thank members of Dr. John Hammer's laboratory (NHLBI), Drs. Anjelika Gasilina and Dillon Schrock, for their assistance with the 4D-Nucleofector apparatus.

## Author contributions

**Pablo S Contreras**: Conceptualization; Data curation; Formal analysis; Investigation; Writing—review and editing. **José A Martina**: Conceptualization; Data curation; Formal analysis; Investigation; Writing—review and editing. **Katie Rollins**: Conceptualization; Data curation; Formal analysis; Investigation; Writing—review and editing. **Eutteum Jeong**: Data curation; Formal analysis; Investigation; Writing—review and editing. **Alberto Rissone**: Conceptualization; Data curation; Formal analysis; Writing—review and editing. **Rosa Puertollano**: Conceptualization; Supervision; Funding acquisition; Investigation; Writing—original draft.

Source data underlying figure panels in this paper may have individual authorship assigned. Where available, figure panel/source data authorship is listed in the following database record: biostudies:S-SCDT-10_1038-S44319-025-00659-3.

## Funding

## Disclosure and competing interests statement

The authors declare no competing interests.

# Expanded View Figures

**Figure EV1.  Expression and activation of TFE3 isoforms in response to stress.**

(**A**) Immunofluorescence confocal microscopy of ARPE19 cells showing the subcellular distribution of TFE3 (red) in response to treatment with EBSS for 1, 3, 8, 12, and 24 h. Scale bars: 10 μm. (**B**) Immunoblot analysis of protein lysates from HEK293T cells treated with EBSS for 1, 3, 8, 12, and 24 h. (**C**) Quantification of protein levels showing TFE3-L/TFE3-S ratio expressed as fold change as shown in (**B**). Data are presented as mean ± SD of four independent experiments. (ns) not significant [a]$P = 0.9303$; (ns) not significant [b]$P = 0.0981$; **[c]$P = 0.0016$; ****[d]$P < 0.0001$; ****[e]$P < 0.0001$ (one-way ANOVA followed by Dunnett's multiple comparison post-test). (**D**) Quantification of protein levels showing TFE3-L/GAPDH ratio expressed as fold change as shown in (**B**). Data are presented as mean ± SD of four independent experiments. (ns) not significant [a]$P = 0.9980$; (ns) not significant [b]$P = 0.8448$; *[c]$P = 0.0343$; *[d]$P = 0.0315$; **[e]$P = 0.0087$ (one-way ANOVA followed by Dunnett's multiple comparison post-test). (**E**) Immunoblot analysis of protein lysates from ARPE19 cells transiently expressing either active or inactive Rag heterodimers for 24 h and treated with 1 μM MLN4924 for 8 h. (**F**) Quantification of protein levels showing TFE3-L/TFE3 ratio expressed as fold change as shown in (**E**). Data are presented as mean ± SD of four independent experiments. ***[a]$P = 0.0005$; ***[b]$P = 0.0002$; ****[c]$P < 0.0001$ (one-way ANOVA followed by Dunnett's multiple comparison post-test). (**G**) Immunoblot analysis of protein lysates from HEK293T cells transiently expressing either active or inactive Rag heterodimers for 24 h. (**H**) Quantification of protein levels showing TFE3-L/TFE3 ratio expressed as fold change as shown in (**G**). Data are presented as mean ± SD of three independent experiments. **$P = 0.0027$ (unpaired Student's *t* test). (**I**) Immunofluorescence confocal microscopy of RAW 264.7 cells showing the subcellular distribution of TFE3 (green) in response to treatment with 1 μg/ml LPS for 6, 12, 24, and 48 h. Scale bars: 10 μm. (**J**) Immunoblot analysis of protein lysates from RAW 264.7 cells treated with 1 μg/ml LPS for 6, 12, 24, and 48 h.

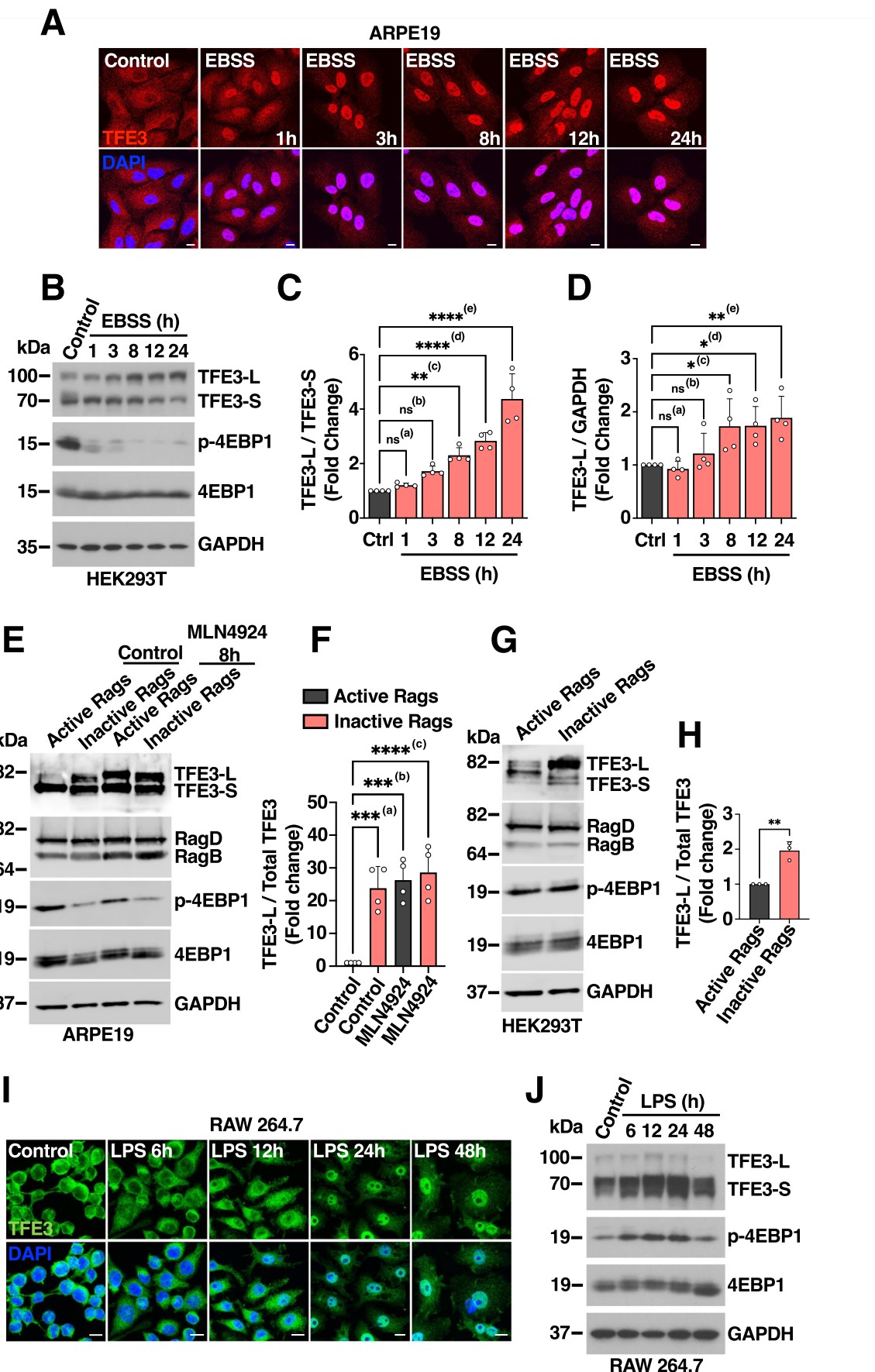

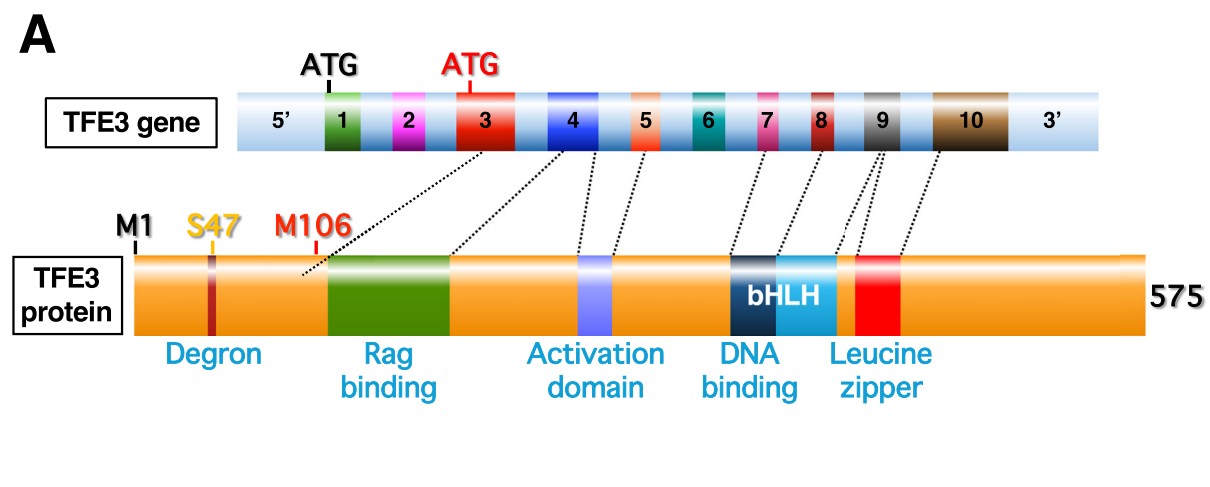

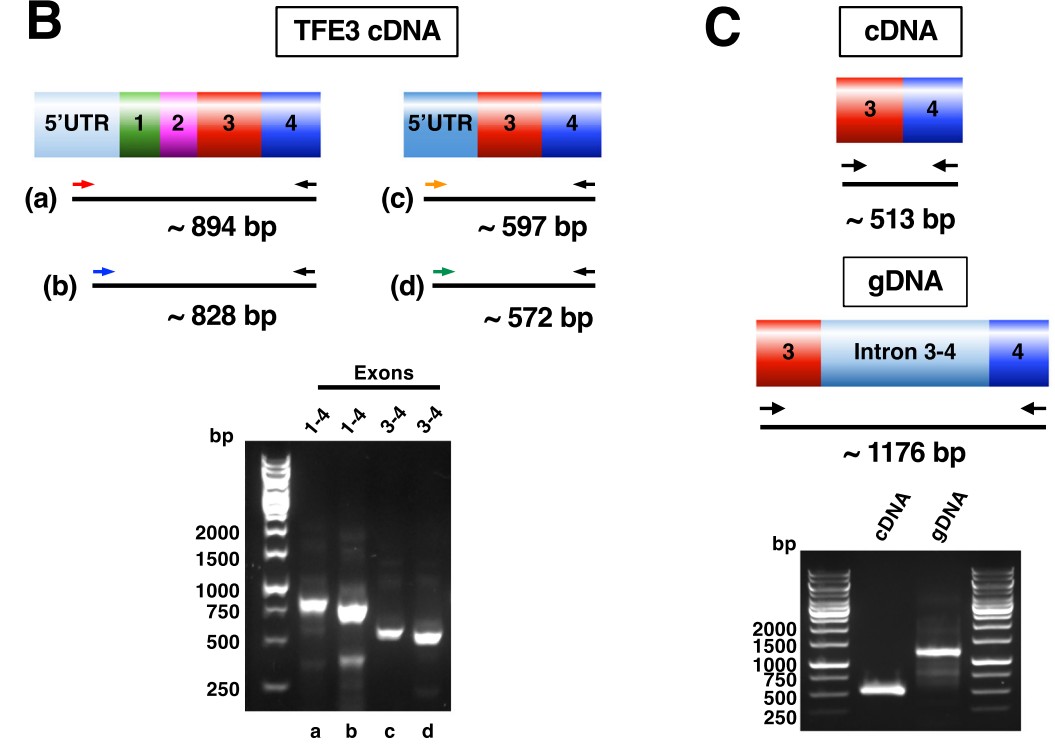

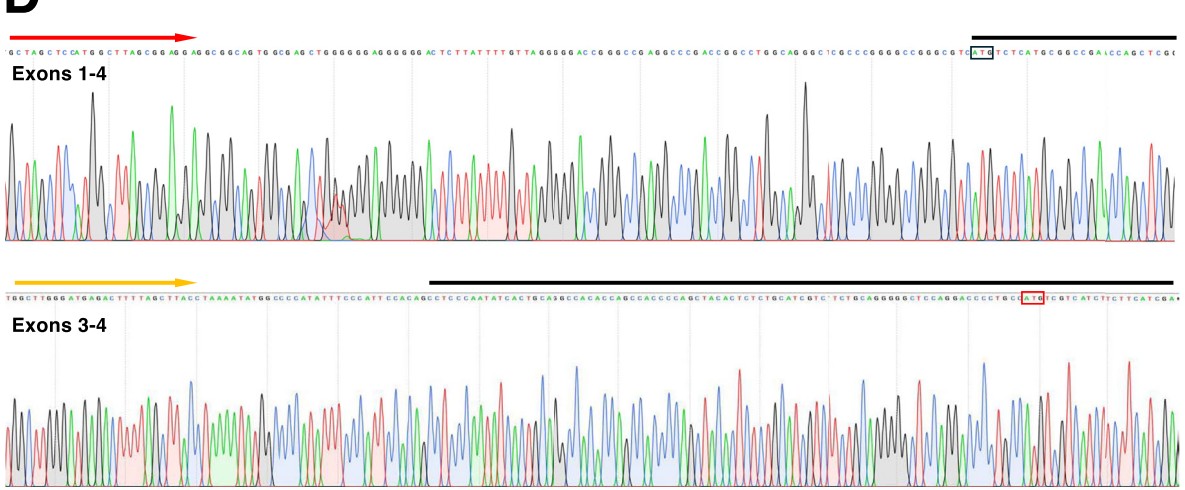

**Figure EV2. TFE3-L and TFE3-S are products of two distinct TFE3 transcripts.**

(A) Schematic diagram of the genomic structure of TFE3 representing exons 1 through 10, their corresponding intronic regions and the position of two start codons. The dotted lines indicate the region of exons encoding for the conserved functional domains of TFE3 protein. Intron sizes are not to scale, and exons and protein domains sizes and boundaries are approximate. (B) Gel electrophoresis analysis showing the RT-PCR amplified cDNA fragments of TFE3. Black arrows represent a common reverse primer targeting exon 4, red and blue arrows represent forward primers that target the 5′UTR of TFE3-L; orange and green arrows indicate primers to the intronic region upstream TFE3 exon 3. (C) Gel electrophoresis analysis showing amplified PCR fragments of TFE3 from either cDNA or genomic DNA (gDNA) using forward and reverse primers (black arrows) targeting exons 3 and 4, respectively. (D) Partial sequence chromatogram of the subcloned cDNA amplified fragments (a) and (c) shown in (B). Black lines indicate the beginning of the coding sequences for exons 1 and 3, and the start codons corresponding to methionines 1 and 106 are indicated with black and red rectangles, respectively.

 

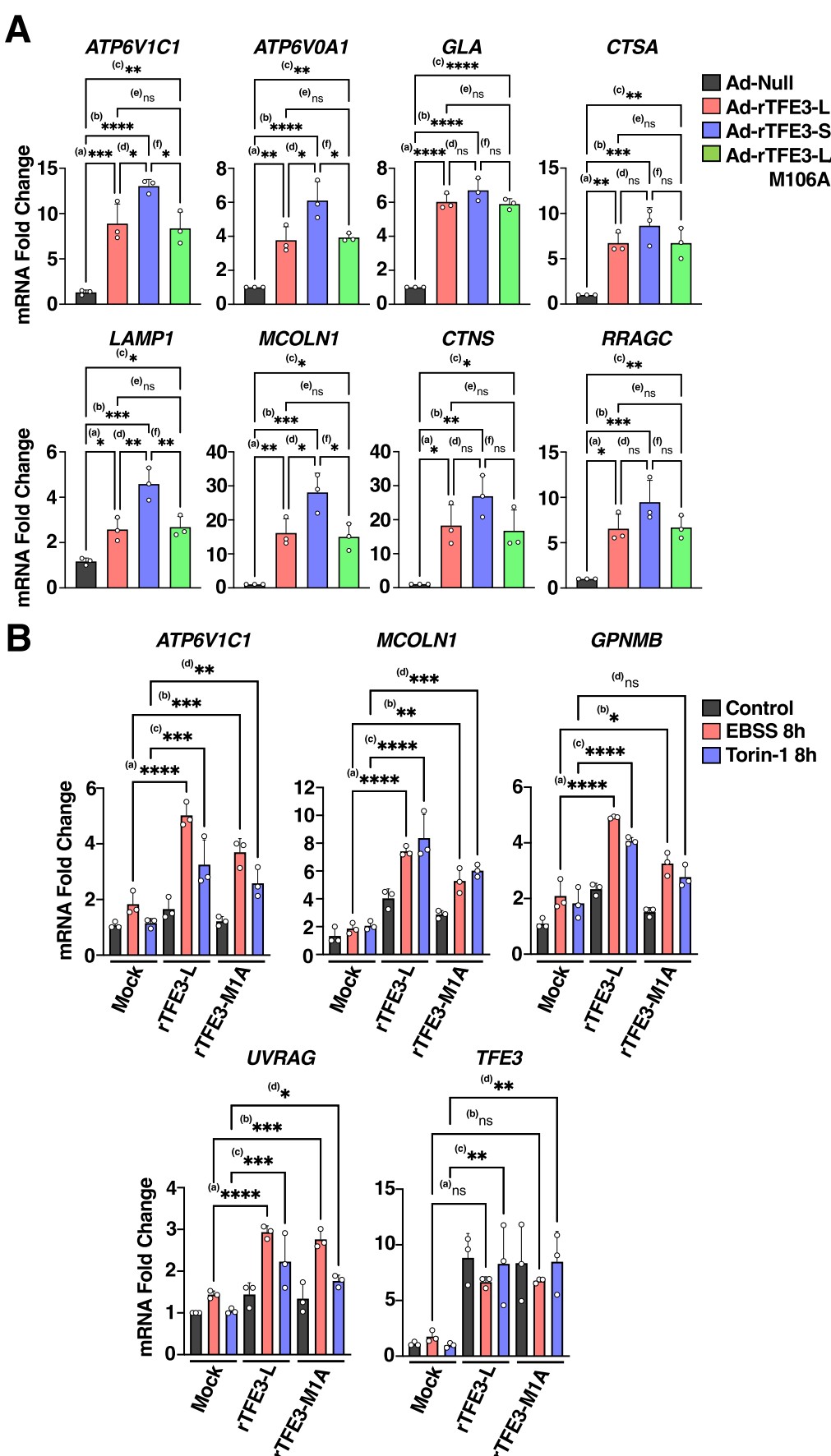

◀ **Figure EV3.  Overexpression of recombinant TFE3-L-Myc, TFE3-S-Myc, and TFE3-L-M106A-Myc induces expression of lysosomal genes.**

(A) Relative quantitative RT-PCR analysis of the mRNA expression of lysosomal genes in ARPE19 cells infected with adenovirus expressing recombinant TFE3-L-Myc, TFE3-S-Myc, TFE3-L/M106A-Myc or control adenovirus (Null) for 30 h. Data are presented as mean ± SD of three independent experiments. ATP6V1C1 ([***][a]$P = 0.0010$; [****][b]$P < 0.0001$; [**][c]$P = 0.0016$; [*][d]$P = 0.0352$; (ns) not significant [e]$P = 0.9696$; [*][f]$P = 0.0192$), ATP6V0A1 ([**][a]$P = 0.0053$; [****][b]$P < 0.0001$; [**][c]$P = 0.0038$; [*][d]$P = 0.0144$; (ns) not significant [e]$P = 0.9923$; [*][f]$P = 0.0207$), GLA ([****][a]$P < 0.0001$; [****][b]$P < 0.0001$; [****][c]$P < 0.0001$; (ns) not significant [d]$P = 0.3036$; (ns) not significant [e]$P = 0.9836$; (ns) not significant [f]$P = 0.1908$), CTSA ([**][a]$P = 0.0049$; [***][b]$P = 0.0008$; [**][c]$P = 0.0049$; (ns) not significant [d]$P = 0.4090$; (ns) not significant [e]$P > 0.9999$; (ns) not significant [f]$P = 0.4057$), LAMP1 ([*][a]$P = 0.0363$; [***][b]$P = 0.0002$; [*][c]$P = 0.0254$; [*][d]$P = 0.0054$; (ns) not significant [e]$P = 0.9936$; [*][f]$P = 0.0074$), MCOLN1 ([**][a]$P = 0.0075$; [***][b]$P = 0.0002$; [*][c]$P = 0.0118$; [*][d]$P = 0.0279$; (ns) not significant [e]$P = 0.9842$; [*][f]$P = 0.0173$), CTNS ([*][a]$P = 0.0164$; [**][b]$P = 0.0014$; [*][c]$P = 0.0266$; (ns) not significant [d]$P = 0.2641$; (ns) not significant [e]$P = 0.9835$; (ns) not significant [f]$P = 0.1642$), RRAGC ([*][a]$P = 0.0113$; [***][b]$P = 0.0008$; [**][c]$P = 0.0099$; (ns) not significant [d]$P = 0.1882$; (ns) not significant [e]$P = 0.9996$; (ns) not significant [f]$P = 0.2160$) (one-way ANOVA followed by Tukey's multiple comparison post-test). (B) Relative quantitative RT-PCR analysis of the mRNA expression of lysosomal and autophagy genes in ARPE19 cells transiently expressing recombinant TFE3-L-Myc or TFE3-L/M1A-Myc for 24 h and treated with either EBSS or 250 nM Torin-1 for 8 h. Data are presented as mean ± SD of three independent experiments. ATP6V1C1 ([****][a]$P < 0.0001$; [**][b]$P = 0.0008$; [****][c]$P = 0.0002$; [**][d]$P = 0.0100$), MCOLN1 ([****][a]$P < 0.0001$; [**][b]$P = 0.0012$; [****][c]$P < 0.0001$; [***][d]$P = 0.0002$), GPNMB ([****][a]$P < 0.0001$; [*][b]$P = 0.0114$; [****][c]$P < 0.0001$; (ns) not significant [d]$P = 0.0526$), UVRAG ([****][a]$P < 0.0001$; [***][b]$P = 0.0001$; [***][c]$P = 0.0005$; [*][d]$P = 0.0433$), TFE3 ((ns) not significant [a]$P = 0.0649$; (ns) not significant [b]$P = 0.0560$; [**][c]$P = 0.0026$; [**][d]$P = 0.0020$) (two-way ANOVA followed by Tukey's multiple comparison post-test).

  

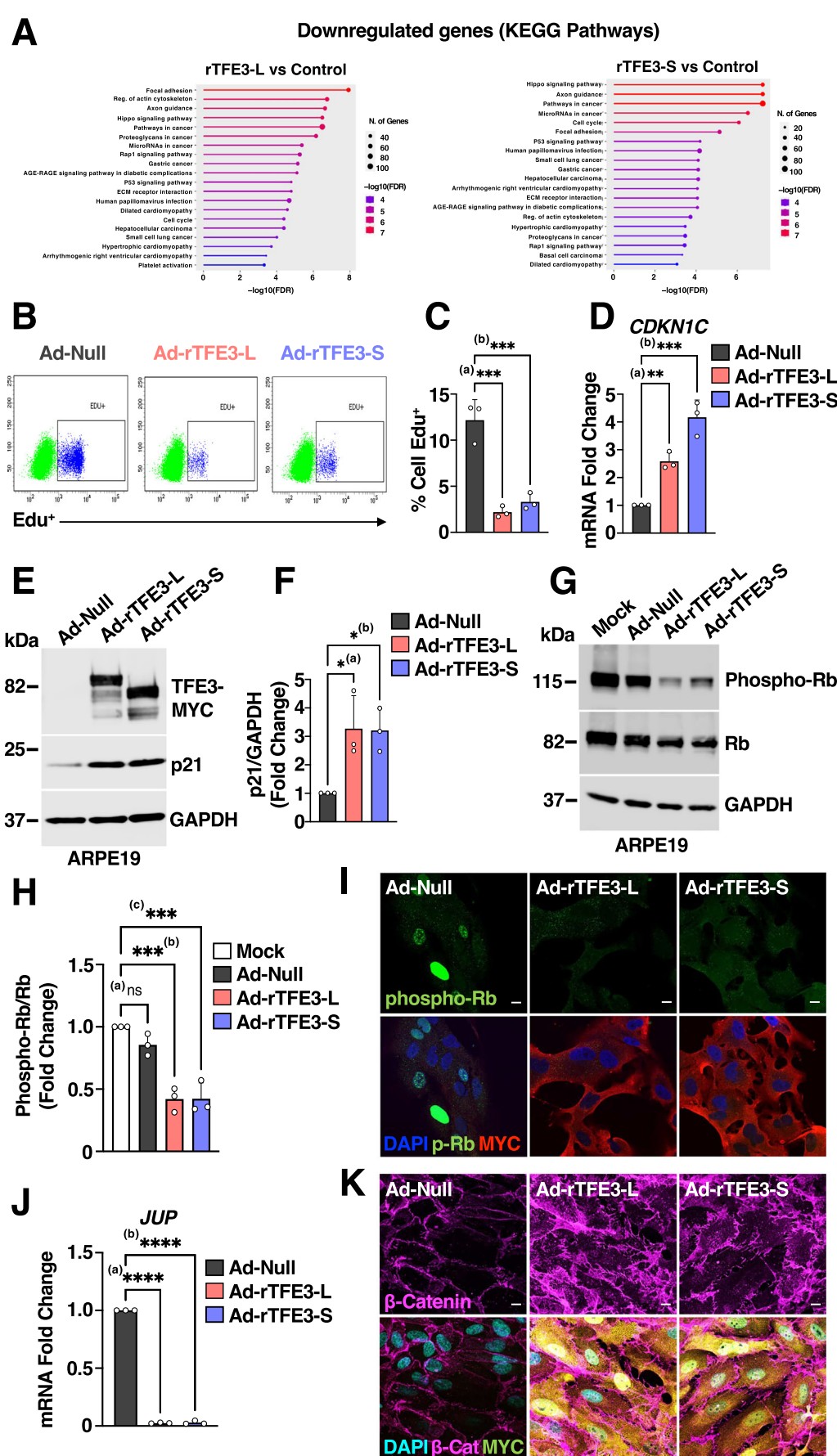

**Figure EV4. TFE3-L-Myc and TFE3-S-Myc overexpression reduces proliferation and alters cell–cell contacts.**

(A) KEGG pathway enrichment analysis of genes downregulated by TFE3-L-Myc or TFE3-S-Myc. Circle size represents the number of genes, while the color scale indicates the -log10 of the false discovery rate (FDR). (B) Flow cytometry analysis of 5-ethynyl-2'-deoxyuridine (EdU) incorporation in ARPE19 cells infected with control adenovirus (Null) or adenovirus expressing recombinant TFE3-L-Myc and TFE3-S-Myc for 30 h. (C) Quantification of the percentage of infected ARPE19 cells with incorporated EdU as shown in (B). Data are presented as mean ± SD of three independent experiments. $^{***(a)}P = 0.0003$; $^{***(b)}P = 0.0005$ (one-way ANOVA followed by Dunnett's multiple comparison post-test). (D) Relative quantitative RT-PCR analysis of the mRNA expression of CDKN1C gene in ARPE19 cells infected with adenovirus expressing recombinant TFE3-L-Myc, TFE3-S-Myc or Control for 30 h. Data are presented as mean ± SD of three independent experiments. $^{**(a)}P = 0.0052$; $^{***(b)}P = 0.0001$ (one-way ANOVA followed by Dunnett's multiple comparison post-test). (E) Immunoblot analysis of protein lysates from ARPE19 cells infected with adenovirus expressing recombinant TFE3-L-Myc, TFE3-S-Myc, or control adenovirus (Null) for 30 h. (F) Quantification of protein levels showing p21/GAPDH ratio expressed as fold change as shown in (E). Data are presented as mean ± SD of three independent experiments. $^{*(a)}P = 0.0234$; $^{*(b)}P = 0.0262$ (one-way ANOVA followed by Dunnett's multiple comparison post-test). (G) Immunoblot analysis of protein lysates from ARPE19 cells infected with adenovirus expressing recombinant TFE3-L-Myc, TFE3-S-Myc, or control adenovirus (Null) for 30 h. (H) Quantification of protein levels showing phospho-Rb/Rb ratio expressed as fold change as shown in (G). Data are presented as mean ± SD of three independent experiments. (ns) not significant $^{(a)}P = 0.2001$; $^{***(b)}P = 0.0001$; $^{***(c)}P = 0.0001$ (one-way ANOVA followed by Dunnett's multiple comparison post-test). (I) Immunofluorescence confocal microscopy of ARPE19 cells infected with adenovirus expressing recombinant TFE3-L-Myc, TFE3-S-Myc, or control adenovirus (Null) for 30 h, showing the cellular distribution of phospho-Rb (green) and recombinant TFE3-L-Myc or TFE3-S-Myc (red). Scale bars: 10 μm. (J) Relative quantitative RT-PCR analysis of the mRNA expression of *JUP* in ARPE19 cells infected with adenovirus expressing recombinant TFE3-L-Myc, TFE3-S-Myc, or control adenovirus (Null) for 30 h. Data are presented as mean ± SD of three independent experiments. $^{****(a)}P < 0.0001$; $^{****(b)}P < 0.0001$ (one-way ANOVA followed by Dunnett's multiple comparison post-test). (K) Immunofluorescence confocal microscopy of ARPE19 cells infected with the indicated adenovirus for 30 h, showing cellular distribution of β-Catenin (pseudo-color magenta) and recombinant TFE3-L-Myc or TFE3-S-Myc (green). Scale bars: 10 μm.

 

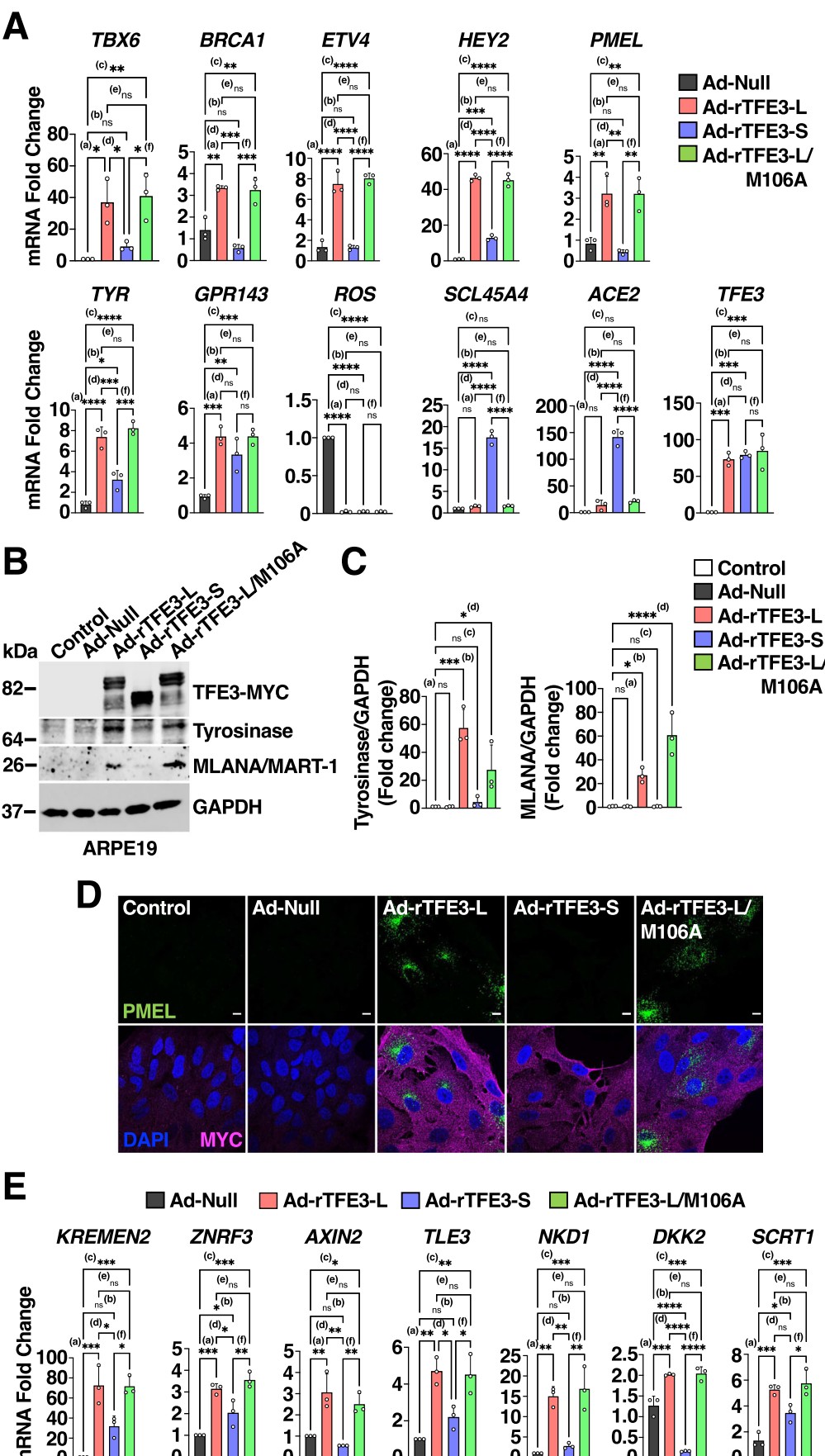

◀ **Figure EV5. Differential gene expression upon overexpression of TFE3 isoforms.**

(A) Relative quantitative RT-PCR analysis of the mRNA expression of the indicated genes in ARPE19 cells infected with adenovirus expressing recombinant TFE3-L-Myc, TFE3-S-Myc, TFE3-L/M106A-Myc or control adenovirus (Null) for 30 h. Data are presented as mean ± SD of three independent experiments. TBX6 ([a]$P = 0.0110$; (ns) not significant [b]$P = 0.7678$; [c]$P = 0.0060$; [d]$P = 0.0412$; (ns) not significant [e]$P = 0.9619$; [f]$P = 0.0213$), BRCA1 ([a]$P = 0.0017$; (ns) not significant [b]$P = 0.1346$; [c]$P = 0.0024$; [d]$P = 0.0001$; (ns) not significant [e]$P = 0.9899$; [f]$P = 0.0002$), ETV4 ([a]$P < 0.0001$; (ns) not significant [b]$P = 0.9998$; [c]$P < 0.0001$; [d]$P < 0.0001$; (ns) not significant [e]$P = 0.7658$; [f]$P < 0.0001$), HEY2 ([a]$P < 0.0001$; [b]$P = 0.0005$; [c]$P < 0.0001$; [d]$P < 0.0001$; (ns) not significant [e]$P = 0.8636$; [f]$P < 0.0001$), PMEL ([a]$P = 0.0049$; (ns) not significant [b]$P = 0.8388$; [c]$P = 0.0050$; [d]$P = 0.0018$; (ns) not significant [e]$P = 0.5424$; [f]$P = 0.0019$), TYR ([a]$P < 0.0001$; [b]$P = 0.0179$; [c]$P < 0.0001$; [d]$P = 0.0006$; (ns) not significant [e]$P = 0.7658$; [f]$P = 0.0002$), GPR143 ([a]$P = 0.0004$; [b]$P = 0.0046$; [c]$P = 0.0004$; (ns) not significant [d]$P = 0.2124$; (ns) not significant [e]$P > 0.9999$; (ns) not significant [f]$P = 0.2084$), ROS ([a]$P < 0.0001$; [b]$P < 0.0001$; [c]$P < 0.0001$; (ns) not significant [d]$P > 0.9999$; (ns) not significant [e]$P = 0.9868$; (ns) not significant [f]$P = 0.9868$), SCL45A4 ((ns) not significant [a]$P = 0.8487$; [b]$P < 0.0001$; (ns) not significant [c]$P = 0.7804$; [d]$P < 0.0001$; (ns) not significant [e]$P = 0.9989$; [f]$P < 0.0001$), ACE2 ((ns) not significant [a]$P = 0.3170$; [b]$P < 0.0001$; (ns) not significant [c]$P = 0.0831$; [d]$P < 0.0001$; (ns) not significant [e]$P = 0.7610$; [f]$P < 0.0001$), TFE3 ([a]$P = 0.0006$; [b]$P = 0.0004$; [c]$P = 0.0002$; (ns) not significant [d]$P = 0.9410$; (ns) not significant [e]$P = 0.7137$; (ns) not significant [f]$P = 0.9535$) (one-way ANOVA followed by Tukey's multiple comparison post-test). (B) Immunoblot analysis of protein lysates from ARPE19 cells infected with adenovirus expressing recombinant TFE3-L-Myc, TFE3-S-Myc, TFE3-L/M106A-Myc or control adenovirus (Null) for 30 h. (C) Quantification of protein levels showing Tyrosinase/GAPDH and MLANA/GAPDH ratios expressed as fold change as shown in (B). Data are presented as mean ± SD of three independent experiments. Tyrosinase ((ns) not significant [a]$P > 0.9999$; [b]$P = 0.0002$; (ns) not significant [c]$P = 0.9798$; [d]$P = 0.0291$), MLANA ((ns) not significant [a]$P > 0.9999$; [b]$P = 0.0115$; (ns) not significant [c]$P > 0.9999$; [d]$P < 0.0001$) (one-way ANOVA followed by Dunnett's multiple comparison post-test). (D) Immunofluorescence confocal microscopy of ARPE19 cells infected with the indicated adenovirus for 30 h, showing the cellular distribution of PMEL (green) and recombinants TFE3-Myc (pseudo-color magenta). Scale bars: 10 μm. (E) Relative quantitative RT-PCR analysis of the mRNA expression of negative regulators of Wnt pathway genes (*KREMEN2, ZNRF3, AXIN2, TLE3, NKD1, DKK2,* and *SCRT1*) in ARPE19 cells infected with adenovirus expressing recombinant TFE3-L-Myc, TFE3-S-Myc, TFE3-L/M106A-Myc or control adenovirus (Null) for 30 h. Data are presented as mean ± SD of three independent experiments. KREMEN2 ([a]$P = 0.0004$; (ns) not significant [b]$P = 0.0515$; [c]$P = 0.0004$; [d]$P = 0.0128$; (ns) not significant [e]$P = 0.9998$; [f]$P = 0.0143$), ZNRF3 ([a]$P = 0.0005$; [b]$P = 0.0367$; [c]$P = 0.0001$; [d]$P = 0.0274$; (ns) not significant [e]$P = 0.5627$; [f]$P = 0.0047$), AXIN2 ([a]$P = 0.0043$; (ns) not significant [b]$P = 0.6984$; [c]$P = 0.0251$; [d]$P = 0.0012$; (ns) not significant [e]$P = 0.5537$; [f]$P = 0.0059$), TLE3 ([a]$P = 0.0011$; (ns) not significant [b]$P = 0.2473$; [c]$P = 0.0015$; (ns) not significant [d]$P = 0.0122$; (ns) not significant [e]$P = 0.9882$; [f]$P = 0.0185$), NKD1 ([a]$P = 0.0017$; (ns) not significant [b]$P = 0.8730$; [c]$P = 0.0008$; [d]$P = 0.0042$; (ns) not significant [e]$P = 0.8692$; [f]$P = 0.0017$), DKK2 ([a]$P = 0.0009$; [b]$P < 0.0001$; [c]$P = 0.0007$; [d]$P < 0.0001$; (ns) not significant [e]$P = 0.9964$; [f]$P < 0.0001$), SCRT1 ([a]$P = 0.0006$; [b]$P = 0.0251$; [c]$P = 0.0003$; (ns) not significant [d]$P = 0.0543$; (ns) not significant [e]$P = 0.8366$; [f]$P = 0.0171$) (one-way ANOVA followed by Tukey's multiple comparison post-test).

 