## [Peer Review File · EMBO Reports]

Differential contribution of TFE3 isoforms to cell motility and invasion

Pablo Contreras, José Martina, Katie Rollins, Eutteum Jeong, Alberto Rissone, and Rosa Puertollano

Corresponding author(s): Rosa Puertollano (puertolr@nhlbi.nih.gov)

Review Timeline:

Submission Date:	15th May 25
Editorial Decision:	26th Jun 25
Revision Received:	5th Sep 25
Editorial Decision:	17th Oct 25
Revision Received:	13th Nov 25
Accepted:	19th Nov 25

Transaction Report:

Dear Dr. Puertollano

Thank you for the submission of your research manuscript to our journal. We have now received the full set of referee reports that is copied below.

As you will see, the referees acknowledge that the findings are interesting and that the conclusions are overall supported by the data presented but they also raise a number of concerns and have suggestions how to further strengthen the study. Referee 1 suggests to test a few key findings in an additional cell line to see whether the roles of the two TFE3 isoforms can be generalized or are cell-type specific. Referee 2 asks for further insight into the regulation by mTORC1. Further insight into the mechanism of isoform-specific transcriptional activity would certainly be of interest and could certainly be addressed at the level of individual target genes.

Given the constructive and supportive comments, we would like to invite you to revise your manuscript with the understanding that the referee concerns (as detailed above and in their reports) must be fully addressed and their suggestions taken on board. Please address all referee concerns in a complete point-by-point response. Acceptance of the manuscript will depend on a positive outcome of a second round of review. It is EMBO Reports policy to allow a single round of revision only and acceptance or rejection of the manuscript will therefore depend on the completeness of your responses included in the next, final version of the manuscript.

We realize that it is difficult to revise to a specific deadline. In the interest of protecting the conceptual advance provided by the work, we recommend a revision within 3 months (September 26th). Please discuss the revision progress ahead of this time with the editor if you require more time to complete the revisions.

I am also happy to discuss the revision further via e-mail or a video call, if you wish.

=====
IMPORTANT NOTE:

We perform an initial quality control of all revised manuscripts before re-review. Your manuscript will FAIL this control and the handling will be delayed IN CASE the following APPLIES:

- 1) A data availability section providing access to data deposited in public databases is missing. If you have not deposited any data, please add a sentence to the data availability section that explains that.
- 2) Your manuscript contains statistics and error bars based on $n=2$. Please use scatter blots in these cases. No statistics should be calculated if $n=2$.

=====

- 2) individual production quality figure files as .eps, .tif, .jpg (one file per figure). Please download our Figure Preparation Guidelines (figure preparation pdf) from our Author Guidelines pages <https://www.embopress.org/page/journal/14693178/authorguide> for more info on how to prepare your figures.

4) a complete author checklist, which you can download from our author guidelines (<https://www.embopress.org/page/journal/14693178/authorguide>). Please insert information in the checklist that is also reflected in the manuscript. The completed author checklist will also be part of the RPF.

5) Please note that all corresponding authors are required to supply an ORCID ID for their name upon submission of a revised manuscript (<https://orcid.org/>). Please find instructions on how to link your ORCID ID to your account in our manuscript tracking system in our Author guidelines (<https://www.embopress.org/page/journal/14693178/authorguide#authorshipguidelines>)

6) We replaced Supplementary Information with Expanded View (EV) Figures and Tables that are collapsible/expandable online. A maximum of 5 EV Figures can be typeset. EV Figures should be cited as 'Figure EV1, Figure EV2' etc... in the text and their respective legends should be included in the main text after the legends of regular figures.

- The "Data Availability" section needs to be moved to the end of the Methods and follow the suggested wording: "The [structural coordinates | microarray | mass spectrometry] data from this publication have been deposited to the [name of the database] database [URL] and assigned the identifier [accession | permalink | hashtag].".

Additional information on source data and instruction on how to label the files are available <https://www.embopress.org/page/journal/14693178/authorguide#sourcedata>

10) Figure legends and data quantification:
The following points must be specified in each figure legend:

- the name of the statistical test used to generate error bars and P values,
- the EXACT p-values,
- the number (n) of independent experiments (please specify technical or biological replicates) underlying each data point,
- the nature of the bars and error bars (s.d., s.e.m.)
- If the data are obtained from n {less than or equal to} 5, show the individual data points in addition to the SD or SEM.
- If the data are obtained from n {less than or equal to} 2, use scatter blots showing the individual data points.

See also the guidelines for figure legend preparation: <https://www.embopress.org/page/journal/14693178/authorguide#figureformat>

11) Our journal encourages inclusion of *data citations in the reference list* to directly cite datasets that were re-used and obtained from public databases. Data citations in the article text are distinct from normal bibliographical citations and should directly link to the database records from which the data can be accessed. In the main text, data citations are formatted as follows: "Data ref: Smith et al, 2001" or "Data ref: NCBI Sequence Read Archive PRJNA342805, 2017". In the Reference list, data citations must be labeled with "[DATASET]". A data reference must provide the database name, accession number/identifiers and a resolvable link to the landing page from which the data can be accessed at the end of the reference. Further instructions are available at <https://www.embopress.org/page/journal/14693178/authorguide#referencesformat>.

12) All Materials and Methods need to be described in the main text using our 'Structured Methods' format. According to this format, the Methods section includes a Reagents and Tools Table (listing key reagents, experimental models, software and relevant equipment and including their sources and relevant identifiers) followed by a Methods and Protocols section describing the methods, ideally using a step-by-step protocol format. The aim is to facilitate adoption of the methodologies across labs. Please download and fill our Reagents and Tools Table template (.docx), which you can find in our author guidelines: <https://www.embopress.org/page/journal/14693178/authorguide#structuredmethods>.

13) As part of the EMBO publication's Transparent Editorial Process, EMBO Reports publishes online a Review Process File to accompany accepted manuscripts. This File will be published in conjunction with your paper and will include the referee reports, your point-by-point response and all pertinent correspondence relating to the manuscript.

Yours sincerely,

=====

Referee #1:

Contreras, Martina et al. have explored the structural and functional differences between two isoforms of the transcription factor TFE3: the long isoform (TFE3-L) and the short (TFE3-S). The authors provide evidence that the isoforms arise via alternative promoter usage and differ in their stability, regulation and biological function. While both isoforms activate lysosomal and autophagic genes downstream of mTORC1, TFE3-L may have a more important role in promoting cell motility and invasion under chronic stress conditions and in TSC2-deficient cells. The study is well designed and offers mechanistic insight into TFE3 isoform biology.

While the authors present innovative data and mechanistically interesting hypotheses with clinical implications, several conclusions overreach the supporting data. Key mechanistic assumptions (e.g. lysosomal localization, selective promoter usage, and invasive behavior) require further validation. The use of only one cell line, lack of consistent time points for stress assays and overreliance on overexpression systems somewhat weaken the conclusions. Clarifications in methods, presentation and interpretation would improve the manuscript's scientific impact.

Major comments

1. The use of ARPE19 cells exclusively as the experimental model limits the generalizability of the findings. TFE3 function is highly context dependent. Inclusion of at least one other cell line would strengthen conclusions about isoform function. Do all cell lines express both isoforms?

2. The statement (page 6) that "most of the active TFE3 found in the nucleus ... corresponds to the short TFE3 form" (Fig. 1L) does not seem well supported by the data.

3. The data in Figure 1L lack complete temporal analysis. While starvation is examined over a time course (Fig. 1A), other stressors (NaAsO₂, LPS, CHQ) are evaluated only at two time points. Additional time-resolved experiments would be helpful to validate isoform stability and nuclear translocation, although not essential.

The comment that TFE3 isoforms localize to lysosomes (page 8) after Torin-1 treatment is speculative. No lysosomal markers are provided in Figure 2E. This statement should be revised or supported with additional experiments.

4. The CRISPR knockout design of TFE3-L is not well described. Given the shared sequence between TFE3 isoforms, it is essential to explain how isoform specificity was achieved. It is interesting that TFE3S levels are higher in the TFE3L KO cells. What do the authors think is the mechanism of this?

5. The abstract and discussion should avoid implying causal roles in cancer metastasis based on scratch and invasion assays, without in vivo or clinical correlation.

On page 7/8, when discussing Figure 2B, the authors say that TFE3S is translocated to the nucleus with Torin1, but it appears from the data that TFE3L behaves similarly. The next sentence says that the behavior of TFE3L "indicating" Rag- and 14-3-3-retention, but since this has not been shown experimentally another word choice would be better (perhaps "consistent with").

6. On page 15, the manuscript cites Kwiatkowski et al. (2016) and Zhu et al. (2023) to support increased metastasis in TSC2-deficient tumors. However, Kwiatkowski et al. does not demonstrate such findings, and tumors in TSC are predominantly benign and Zhu et al. is a literature review and not a research study. This statement that the current data are "in agreement with prior studies ..." should probably be removed.

7. One of the most interesting pieces of data is Figure 6E, in which elevated invasion is seen in TSC2 KO cells, and less invasion in the TFEL KO, despite what appears to be a higher level of TFE3S. It would be interesting to know if TFE3L KO also decreases invasion in the WT cells, but it is recognized that this is likely beyond the scope of this work.

Minor comments:

1. In Figure 2E the merged images are incorrectly labeled as 'DAPI'. Each channel should be shown separately to make this easier to interpret.

2. In Figures 1L vs. 2B: treatment durations in Torin-1 and stress experiments should be clarified in the figure legends.

Referee #2:

In this manuscript, Contreras et al. investigate the functional differences between two isoforms of TFE3, TFE3-S (short) and TFE3-L (long), which are proposed to be generated from alternative promoters. They show that TFE3-L, but not TFE3-S, accumulates in response to stress conditions leading to mTORC1 inactivation, due to a previously described phospho-degron contained in its unique N-terminal region. Furthermore, they suggest that TFE3-L, while sharing much of the transcriptional activity of TFE3-S, uniquely induces genes involved in cell migration and invasion, conferring pro-invasive properties. This is a potentially interesting manuscript, as it addresses a longstanding question about the specific roles of two well-known TFE3 isoforms. However, while the study provides several intriguing observations, it remains largely descriptive, and several mechanistic aspects remain elusive. Furthermore, some of the main conclusions are not supported by the data, thus requiring more robust experimental validation.

Specific points:

1. Identity and sequence of TFE3 isoforms

To characterize the two alternative isoforms, the authors performed RT-PCR analyses using forward primers in exon 1 (for TFE3-L) or intron 3 (upstream of the putative alternative ATG of TFE3-S), and a common reverse primer in exon 4. However, no sequence analyses downstream of exon 4 were performed, based on the assumption that both isoforms are identical in the C-terminal region and encode full-length functional proteins. This assumption is not sufficiently supported. To establish the existence and sequence of TFE3-L and TFE3-S, the authors should amplify, via RT-PCR, the full-length sequences, using isoform-specific forward primers and a reverse primer targeting the 3' UTR or stop codon. PCR products should then be cloned (e.g. using TA cloning) and several clones should be sequenced. This approach will allow determining: a) if the full-length isoforms are expressed; 2) if additional splicing or sequence variations occur in the central coding region; 3) if additional isoforms are also expressed.

2. Alternative promoter usage

The section on page 6 entitled "TFE3-L and TFE3-S are generated by the use of alternative promoters", is not supported by direct experimental evidence. The authors hypothesize the use of alternative promoters based on different transcription start sites, but do not identify or functionally validate specific promoter regions. To support the existence of distinct promoters, the authors should perform a systematic analysis of each promoter, by analyzing available CAGE or ENCODE data to support promoter predictions, mapping promoter activity using luciferase reporter assays, or assessing H3K4me3 and RNA Pol II to confirm active transcriptional initiation sites. If such analyses are not feasible at this stage, the section title and interpretation

should be revised.

3. Regulation by mTORC1

The authors propose that mTORC1 regulates TFE3-L stability via phosphorylation at the N-terminal degron. Supporting this, the authors show that nutrient starvation and expression of inactive Rag GTPases promotes TFE3-L accumulation. However, in Fig. 1L the use of Torin, a potent mTORC1 inhibitor, does not result in TFE3-L increased levels, which appears inconsistent with mTORC1 dependency. How do the authors explain these contradictory results? In Fig. 1C-D, the authors state that a rapid mTORC1 reactivation is followed by a decrease in TFE3-L levels. Does refeeding in the presence of Torin prevent TFE3-L degradation? Similarly, does Torin stabilize TFE3-L in cells overexpressing active Rags (Fig. EV1D)? Can active Rags suppress TFE3-L accumulation during starvation, and is this suppression reversed by Torin? Clarifying these points is crucial to support the conclusion that mTORC1 regulates TFE3-L levels.

4. Non-canonical mTORC1 signaling

On page 6, the authors state: "accumulation of TFE3-L was only observed under conditions in which mTORC1 was inactivated for prolonged periods of time (EBSS and Torin-1 treatment)", referring to 4E-BP1 phosphorylation as a readout for mTORC1 inhibition. Recent evidence has shown that TFEB and TFE3 are regulated by a non-canonical mTORC1 signaling pathway that is specifically regulated by the FLCN complex (PMID: 32612235, 35358174). Notably, the authors' own data support that TFE3-L is specifically regulated by this pathway, showing that FLCN depletion promotes TFE3-L accumulation without affecting 4E-BP1 phosphorylation (Fig. 1G). Conversely, short-term treatment with LLOMe, which has been shown to activate TFEB by specifically inhibiting non-canonical mTORC1 signaling (PMID: 32989250), does not affect TFE3-L levels (Fig. 1L). These data highlight the need for a better mechanistic understanding of mTORC1-mediated regulation of TFE3-L. Does prolonged LLOMe treatment induce TFE3-L accumulation? Does expression of a constitutively active form of RagC rescue this phenotype? Is expression of constitutively inactive RagC sufficient for TFE3-L accumulation?

The authors should update the text to acknowledge non-canonical regulation and cite the relevant literature (PMIDs: 32612235, 35358174, 32989250, see also point 7).

5. Mechanism of isoform-specific transcriptional activity

What is the mechanism by which TFE3-L induces the specific expression of certain genes? Does the longer isoform bind to different promoters than TFE3-S? Are TFEB and MITF able to induce TFE3-L-regulated genes? Could the TFE3-L N-terminal region promote the expression of TFE3-L-specific genes when fused to TFEB?

In the discussion, the authors hypothesize that "TFE3-L might indirectly affect expression of specific MITF targets". This hypothesis can be tested using MITF-KO cells.

6. Physiological role of TFE3-L

While the study suggests that TFE3-L accumulation promotes migratory and invasive behavior, this isoform also accumulates in non-transformed cells under stress. What is the physiological relevance of TFE3-L expression in these contexts?

7. Literature citation

The manuscript should provide more comprehensive citation of foundational and recent literature. When TFEB/TFE3 regulation of lysosomal biogenesis and autophagy is introduced in the manuscript, the original studies in which these findings were first reported should be cited (PMID: 19556463, 21617040). Likewise, when introducing TFEB/TFE3 regulation by mTORC1, recent publications elucidating the non-canonical regulation of TFEB and TFE3 via the FLCN complex should also be cited (PMID: 32612235, 35358174, 32989250).

Referee #3:

This article conducts a systematic exploration of the partial redundancy and functional disparities of the two isoforms of TFE3. The authors showed that compared with TFE3-S, TFE3-L is associated with metastatic behaviors, which may hold significant relevance in the context of cancer. The key findings of this study are both interesting and significant. It utilizes a variety of molecular biology approaches and provides compelling evidence, thus facilitating a deeper understanding and broadening the functions of TFE3. However, there are also some inconsistent data within this manuscript. In addition, regarding the effects of TFE3-L on cell migration and invasion compared to TFE3-S, the authors did not explore the underlying mechanisms. It remains to be explored whether such function is related to the autophagy or transcriptional function of TFE3-L. This would contribute to a more profound understanding of the non-canonical functions of TFE3-L and enhance the impact of this article. The following are several suggestions for revisions.

1. In Figure EV1B, subsequent to the EBSS treatment, the level of TFE3-S gradually declined. However, in Figure 1A, TFE3-S remains stable following EBSS starvation. Is this attributable to the different cell types employed? Could the authors repeat this experiment or provide an explanation for such discrepancies?

2. In Figure 1E, the degradation of TFE3-L by MG132 was not prevented. Instead, its degradation was merely delayed when compared to that under the Re-fed condition. Notably, after 6 hours in particular, the level of TFE3-L still decreased significantly.

3. Compared to Figure 1C, in Figure 1E why TFE3-S becomes unstable following the addition of MG132?

4. In Figure 2F, could you please explain the reason why MLN4924, which is supposed to stabilize TFE3-L, reduces the level of TFE3-M1A? But it seems to stabilize both TFE3-L and TFE3-S in Figure 1J.

5. In Figure 3E, to which specific WIPI does the term "WIPI" refer? WIPI2?

6. In Figure 3E, under basal conditions, overexpression of TFE3-S shows better effects on the expression of autophagy-related genes as compared to TFE3-L. However, in Figure EV3B, subsequent to treatment with EBSS or Torin1, TFE3-L demonstrates a more pronounced effect in regulating the expression of autophagy-related genes compared to TFE3-S. Under EBSS starvation conditions, both TFE3-L and TFE3-S translocate to the nucleus to promote the expression of the target genes and the protein level of TFE3-S appears to be much higher than that of TFE3-L. Why would TFE3-L have better effects than TFE3-S on autophagy-related genes expression? This discrepancy should also be discussed in the discussion section.

7. In Figure 4-6, the author investigated the impacts of TFE3-L and TFE3-S on cell proliferation, cell migration, and invasion. TFE3-L demonstrated a much better effect on cell migration and invasion compared with TFE3-S. Does TFE3-L have an impact on the aforementioned phenotypes in the cells with defective autophagy? Are these functions associated with the transcriptional activity of TFE3-L? Is it possible to test the impact of a TFE3-L mutant, which is unable to translocate into the nucleus, on cell migration and invasion?

8. In Figure 6, the authors showed that TFE3-L promoted cell migration and invasion of TSC KO cells. However, this experiment was mainly conducted in HeLa cells. Will there be consistent effects in other tumor cells upon TSC knockout or knockdown?

9. Given that this study is primarily conducted in cultured cells rather than through in vivo investigations, the statement regarding cancer should be made with prudence. Moreover, such limitations should be included in the discussion.

Referee #1:

Contreras, Martina et al. have explored the structural and functional differences between two isoforms of the transcription factor TFE3: the long isoform (TFE3-L) and the short (TFE3-S). The authors provide evidence that the isoforms arise via alternative promoter usage and differ in their stability, regulation and biological function. While both isoforms activate lysosomal and autophagic genes downstream of mTORC1, TFE3-L may have a more important role in promoting cell motility and invasion under chronic stress conditions and in TSC2-deficient cells. The study is well designed and offers mechanistic insight into TFE3 isoform biology.

We thank the reviewer for the support

While the authors present innovative data and mechanistically interesting hypotheses with clinical implications, several conclusions overreach the supporting data. Key mechanistic assumptions (e.g. lysosomal localization, selective promoter usage, and invasive behavior) require further validation. The use of only one cell line, lack of consistent time points for stress assays and overreliance on overexpression systems somewhat weaken the conclusions. Clarifications in methods, presentation and interpretation would improve the manuscript's scientific impact.

Major comments

1. The use of ARPE19 cells exclusively as the experimental model limits the generalizability of the findings. TFE3 function is highly context dependent. Inclusion of at least one other cell line would strengthen conclusions about isoform function. Do all cell lines express both isoforms?

Please note that in the previous version of the manuscript we had shown the presence of the two TFE3 isoforms not only in ARPE-19, but also in HEK293T and HeLa-TSC2-KO cells. To further corroborate our findings, we have also analyzed HeLa and U2OS cells and performed the following experiments: 1) we show a significant and progressive increase in the levels of TFE3-L in HeLa cells following prolonged starvation (Appendix Figure S1A,B); 2) we confirmed the increased ability of TFE3-L to induce expression of specific targets, such as *CHI3L1* and *CITED1*, both in HeLa and U2OS cells (Appendix Figure S9A,B); 3) we report increased motility of HeLa cells expressing TFE3-L when compared with control cells or cells expressing TFE3-S (Appendix Figure S6). Therefore, our findings seem to be consistent in a variety of different cell lines.

2. The statement (page 6) that "most of the active TFE3 found in the nucleus ... corresponds to the short TFE3 form" (Fig. 1L) does not seem well supported by the data.

We apologize if this point was not sufficiently clear. We suggest that TFE3-S is the primary isoform mediating early stress responses, since it is the predominant isoform

present in cells at these early times. For example, we show that TFE3-L is practically absent in ARPE-19 cells starved for 1 or 3h (Figure 1A and 1B). Therefore, the TFE3 accumulated in the nucleus at these times (Figure EV1A) must correspond mostly to TFE3-S. This is also the case for ARPE-19 cells treated with NaAsO₂ for 2 or 4h (Figure 1L and 1M). To further strengthen this idea, we have now performed subcellular fractionation experiments in ARPE-19 cells treated with NaAsO₂, LLOMe, or EBSS for 2h. As seen in Appendix Figure S3A-C, most of the TFE3 present in the nuclear fraction at this time does indeed correspond to TFE3-S.

3. The data in Figure 1L lack complete temporal analysis. While starvation is examined over a time course (Fig. 1A), other stressors (NaAsO₂, LPS, CHQ) are evaluated only at two time points. Additional time-resolved experiments would be helpful to validate isoform stability and nuclear translocation, although not essential.

We thank the reviewer for the suggestion. We have now performed longer incubation times with different stressors. We found that even if with different levels of efficiency, all the tested stressors could induce a significant increase in the levels of TFE3-L (Appendix Figure S4A-C). The only exception was LLOMe, which induced TFE3 nuclear accumulation and TFE3-L stabilization at 12h but not at 24h, a phenomenon that may reflect the ability of cells to eliminate or repair damaged lysosomes, thus decreasing the stress (Appendix Figure S4A-C). Another exception were the data that we previously showed indicating that incubation of RAW 264.7 cells with LPS causes efficient TFE3-S activation but not TFE3-L stabilization (Figure EV1H and EV1I). These results, suggest that TFE3-L levels increase in response to prolonged exposure to a wide variety of stress conditions.

The comment that TFE3 isoforms localize to lysosomes (page 8) after Torin-1 treatment is speculative. No lysosomal markers are provided in Figure 2E. This statement should be revised or supported with additional experiments.

We apologized by the oversight. We have described in many of our previous studies that Torin-1 causes accumulation of TFEB and TFE3 on the lysosomal surface and that this recruitment requires active Rag GTPases (Martina and Puertollano, 2013; Martina et al., 2024; Martina et al., 2021; Tapia et al., 2025). However, the reviewer is correct in that co-localization of our mutants with lysosomal markers must be provided. Please note that co-localization of rTFE3-L, rTFE3-S, and rTFE3-M1A with LAMP1 in ARPE-19 cells treated with Torin-1 is now shown in Appendix Figure S5B.

4. The CRISPR knockout design of TFE3-L is not well described. Given the shared sequence between TFE3 isoforms, it is essential to explain how isoform specificity was achieved.

This information has been added to the methods section. In addition, the guide RNA sequence is now included in Table EV3.

It is interesting that TFE3S levels are higher in the TFE3L KO cells. What do the authors think is the mechanism of this?

We agree with the reviewer in that TFE3-L depletion seems to slightly increase the levels of TFE3-S in western blot assays. This is an intriguing observation that may suggest some sort of negative regulation of the long over the short form in basal conditions. However, other mechanisms are also possible, including as a more general compensatory effect at transcriptional and/or translational level, or just an increased accessibility of the short transcripts to the translation machinery. Future studies will be focused on clarifying the mechanisms at the base of this phenomenon.

5. The abstract and discussion should avoid implying causal roles in cancer metastasis based on scratch and invasion assays, without in vivo or clinical correlation.

The abstract and discussion have now been modified.

On page 7/8, when discussing Figure 2B, the authors say that TFE3S is translocated to the nucleus with Torin1, but it appears from the data that TFE3L behaves similarly.

Yes, this is the expectation, since both isoforms share the same Rag and 14-3-3 binding domains.

The next sentence says that the behavior of TFE3L "indicating" Rag- and 14-3-3- retention, but since this has not been shown experimentally another word choice would be better (perhaps "consistent with").

To further confirm that TFE3-L and TFE3-S share the same mechanism of cytosolic retention, we performed pull-down experiments with either active or inactive Rags. Treatment with MLN4924 for 5h was used to stabilize and accumulate endogenous TFE3-L. As expected, and in agreement with our previous study (Martina et al., 2014), we found that both isoforms interact with active, but not with inactive, Rags (Appendix Figure S5A). In addition, mTOR inhibition by Torin-1 caused dephosphorylation of serine 321 (the residue responsible for the binding to 14-3-3) and nuclear translocation of both TFE3-L and TFE3-S (Appendix Figure S5B,C). These data, together with the fact that both proteins share the same sequence, except for the first 105 amino terminal residues only present in TFE3-L, strongly suggest a common regulatory mechanism of activation.

6. On page 15, the manuscript cites Kwiatkowski et al. (2016) and Zhu et al. (2023) to support increased metastasis in TSC2-deficient tumors. However, Kwiatkowski et al. does not demonstrate such findings, and tumors in TSC are predominantly benign and Zhu et al. is a literature review and not a research study. This statement that the current data are "in agreement with prior studies ..." should probably be removed.

We thank the reviewer for the correction. The indicated references have now been substituted by other studies reporting instances of increased metastasis in TSC tumors (Meredith et al., 2023; Gupta et al., 2024).

7. One of the most interesting pieces of data is Figure 6E, in which elevated invasion is seen in TSC2 KO cells, and less invasion in the TFEL KO, despite what appears to be a higher level of TFE3S. It would be interesting to know if TFE3L KO also decreases invasion in the WT cells, but it is recognized that this is likely beyond the scope of this work.

This is an interesting point but difficult to address due to the very low levels of TFE3-L in WT cells. In contrast, the suggested inability of mTORC1 to efficiently phosphorylate TFE3 in TSC2-KO cells is consistent with the accumulation of TFE3-L and the increased motility observed in these cells.

Minor comments:

1. In Figure 2E the merged images are incorrectly labeled as 'DAPI'. Each channel should be shown separately to make this easier to interpret.

This has been corrected.

2. In Figures 1L vs. 2B: treatment durations in Torin-1 and stress experiments should be clarified in the figure legends.

This information has been added to the figure legends.

Referee #2:

In this manuscript, Contreras et al. investigate the functional differences between two isoforms of TFE3, TFE3-S (short) and TFE3-L (long), which are proposed to be generated from alternative promoters. They show that TFE3-L, but not TFE3-S, accumulates in response to stress conditions leading to mTORC1 inactivation, due to a previously described phospho-degron contained in its unique N-terminal region. Furthermore, they suggest that TFE3-L, while sharing much of the transcriptional activity of TFE3-S, uniquely induces genes involved in cell migration and invasion, conferring pro-invasive properties. This is a potentially interesting manuscript, as it addresses a longstanding question about the specific roles of two well-known TFE3 isoforms.

We thank the reviewer for the support

However, while the study provides several intriguing observations, it remains largely descriptive, and several mechanistic aspects remain elusive. Furthermore, some of the main

conclusions are not supported by the data, thus requiring more robust experimental validation.

Specific points:

1. Identity and sequence of TFE3 isoforms. To characterize the two alternative isoforms, the authors performed RT-PCR analyses using forward primers in exon 1 (for TFE3-L) or intron 3 (upstream of the putative alternative ATG of TFE3-S), and a common reverse primer in exon 4. However, no sequence analyses downstream of exon 4 were performed, based on the assumption that both isoforms are identical in the C-terminal region and encode full-length functional proteins. This assumption is not sufficiently supported. To establish the existence and sequence of TFE3-L and TFE3-S, the authors should amplify, via RT-PCR, the full-length sequences, using isoform-specific forward primers and a reverse primer targeting the 3' UTR or stop codon. PCR products should then be cloned (e.g. using TA cloning) and several clones should be sequenced. This approach will allow determining: a) if the full-length isoforms are expressed; 2) if additional splicing or sequence variations occur in the central coding region; 3) if additional isoforms are also expressed.

To address the reviewer's comment, a set of two different reverse primers targeting the TFE3 gene stop codon region, together with isoform-specific forward primers, were used to amplify the cDNA sequence corresponding to TFE3-L and TFE3-S via RT-PCR (Figure A, attached to this rebuttal letter). The PCR products were sub-cloned into pJET1.2/blunt cloning vector, several bacterial colonies were picked and the plasmids carrying the PCR products were Sanger sequenced. As expected, the analysis of the sequences revealed the presence of both full length TFE3-L and TFE3-S, corroborating the existence of independent TFE3-L and TFE3-S transcripts (Figure A). We also observed in some clones a deletion of 65 bp at the beginning of TFE3-L sequence, which was the product of a cryptic splicing site that introduced a frameshift, generating a premature stop codon and a potential truncated fragment of approximately 109 amino acids. Importantly, we did not detect any nucleotide deletion in TFE3-S cDNA sequences. These data suggest that TFE3-L and TFE3-S are the predominant TFE3 isoforms in ARPE-19 cells.

Finally, we found two sequences at the Human EST database mapping the TFE3-S isoform, confirming our cloning and expression data.

**DA118756 BRACE3 Homo sapiens cDNA clone BRACE3038915 5', mRNA sequence
Tissue: cerebellum (<https://www.ncbi.nlm.nih.gov/nucleotide/DA118756>)
DA458667 CTONG3 Homo sapiens cDNA clone CTONG3004774 5', mRNA sequence
Tissue: tongue, tumor tissue (<https://www.ncbi.nlm.nih.gov/nucleotide/DA458667>)**

2. Alternative promoter usage. The section on page 6 entitled "TFE3-L and TFE3-S are generated by the use of alternative promoters", is not supported by direct experimental evidence. The authors hypothesize the use of alternative promoters based on different

transcription start sites, but do not identify or functionally validate specific promoter regions. To support the existence of distinct promoters, the authors should perform a systematic analysis of each promoter, by analyzing available CAGE or ENCODE data to support promoter predictions, mapping promoter activity using luciferase reporter assays, or assessing H3K4me3 and RNA Pol II to confirm active transcriptional initiation sites. If such analyses are not feasible at this stage, the section title and interpretation should be revised.

We agree with the reviewer in that a deeper analysis is necessary to better support the notion that the two TFE3 forms may be produced by the use of two alternative promoters. As suggested, we used available online databases to further analyze the potential promoter regions in the human *TFE3* gene. We specifically used the UCSC genome browser and focused on the genomic region encompassing Exon 1 and Exon 3. We repeated our analysis in all the different assemblies of the human genome dataset. While the results were similar, we are only showing the analysis using the most recent dataset (hs1 or T2T genome), which is also the most complete assembly. Using the T2T ENCODE track, we compared histone methylation (H3K4me3 and H3K4me1) and acetylation (H3K27ac) profiles in the available cell lines (HAP-1, SJCRH30 and SJSA1). As shown in Figure B (attached to this rebuttal letter), ENCODE ChIP-seq data show significant binding regions (peaks, black boxes) in the selected regions. In general, the entire region present peaks for all the three histone modifications, indicating that it is likely involved in the transcriptional regulation of the gene. Notably, HAP-1 cell lines present a specific H3K4me3 peak right upstream of Exon 3 (red box in the figure) that may potentially represent the alternative promoter for TFE3-S.

We then analyzed the RNA Pol II coverage using the Cistrome database and the WashU Epigenome browser (Figure B). Notably, intronic region upstream of Exon 3 present a specific peak for the Pol II phosphorylated on Serine 5 (POLR2A-pS5), once again suggesting the presence of a promoter region.

Overall, we believe that the data obtained by our bioinformatic analysis are consistent with the presence of two distinct promoters controlling the transcription of TFE3-L and TFE3-S. However, we agree that a full confirmation would require further defining the characteristics of the human *TFE3* promoter using luciferase reporter assays. Therefore, we have now changed the wording in our manuscript from “alternative promoter” to “alternative transcription initiation sites”.

3. Regulation by mTORC1. The authors propose that mTORC1 regulates TFE3-L stability via phosphorylation at the N-terminal degron. Supporting this, the authors show that nutrient starvation and expression of inactive Rag GTPases promotes TFE3-L accumulation. However, in Fig. 1L the use of Torin, a potent mTORC1 inhibitor, does not result in TFE3-L increased levels, which appears inconsistent with mTORC1 dependency. How do the authors explain these contradictory results?

We think that accumulation of TFE3-L is a time dependent process. For example, EBSS treatment also causes a very efficient mTORC1 inactivation (Figure 1A), but it still takes more than 3h to achieve a significant increase in the levels of TFE3-L. To further address

this point, we have now performed long (12h and 24h) incubations with a variety of stressors. As seen in Appendix Figure S4A-C, we observe a significant increase in the TFE3-L/TFE3-S ratio in most cases. Keep in mind that some of these stressors, such as EBSS and Torin-1, cause an efficient mTORC1 inactivation. Other stressors, such as LLOMe, are known to prevent non-canonical mTOR signaling. Therefore, these data are consistent with a mTORC1-dependent phosphorylation of the degnon.

In Fig. 1C-D, the authors state that a rapid mTORC1 reactivation is followed by a decrease in TFE3-L levels. Does refeeding in the presence of Torin prevent TFE3-L degradation?

The suggested experiment is now shown in Appendix Figure S2A-C. As expected, the presence of Torin-1 during refeed prevented TFE3-L degradation.

Similarly, does Torin stabilize TFE3-L in cells overexpressing active Rags (Fig. EV1D)? Can active Rags suppress TFE3-L accumulation during starvation, and is this suppression reversed by Torin? Clarifying these points is crucial to support the conclusion that mTORC1 regulates TFE3-L levels.

Unfortunately, we were unable to achieve a high enough efficiency of transfection with our active Rags constructs to completely block mTORC1 inactivation in response to starvation. The inhibition of TFE3 nuclear translocation in EBSS-treated cells was clearly observed by immunofluorescence, where we can focus in those cells with high level of Rag expression. However, we observed a high number of untransfected cells, which prevent us from detecting a robust mTORC1 inactivation and TFE3-L accumulation by immunoblot.

4. Non-canonical mTORC1 signaling. On page 6, the authors state: "accumulation of TFE3-L was only observed under conditions in which mTORC1 was inactivated for prolonged periods of time (EBSS and Torin-1 treatment)", referring to 4E-BP1 phosphorylation as a readout for mTORC1 inhibition. Recent evidence has shown that TFEB and TFE3 are regulated by a non-canonical mTORC1 signaling pathway that is specifically regulated by the FLCN complex (PMID: 32612235, 35358174). Notably, the authors' own data support that TFE3-L is specifically regulated by this pathway, showing that FLCN depletion promotes TFE3-L accumulation without affecting 4E-BP1 phosphorylation (Fig. 1G). Conversely, short-term treatment with LLOMe, which has been shown to activate TFEB by specifically inhibiting non-canonical mTORC1 signaling (PMID: 32989250), does not affect TFE3-L levels (Fig. 1L). These data highlight the need for a better mechanistic understanding of mTORC1-mediated regulation of TFE3-L. Does prolonged LLOMe treatment induce TFE3-L accumulation?

Please see our response to your previous point. Treatment with LLOMe for 12h does indeed causes a robust TFE3-L accumulation (Appendix Figure S4B,C). Interestingly, TFE3 nuclear translocation and TFE3-L accumulation decreases at 24h treatment (Appendix Figure S4A,B). This may be due to the ability of cells to eliminate or repair damaged lysosomes, thus decreasing the stress.

Does expression of a constitutively active form of RagC rescue this phenotype? Is expression of constitutively inactive RagC sufficient for TFE3-L accumulation?

Please note that accumulation of TFE3-L following expression of inactive Rags in ARPE-19 and HEK293T cells was already shown in the previous version of the manuscript as Figure EV1D-G. These results were further corroborated in our new Rag pull-down experiments shown in Appendix Figure S5A.

The authors should update the text to acknowledge non-canonical regulation and cite the relevant literature (PMIDs: 32612235, 35358174, 32989250, see also point 7).

The indicated references have been included in the revised version of the manuscript.

5. Mechanism of isoform-specific transcriptional activity. What is the mechanism by which TFE3-L induces the specific expression of certain genes? Does the longer isoform bind to different promoters than TFE3-S?

This is an important question. Given that the DNA binding region is the same in the two isoforms, we hypothesize the possibility of interactions between the N-terminal region of TFE3-L and other transcription factors or activators. However, our attempts to identify these potential regulators by immunoprecipitation of TFE3-L and TFE3-S followed by Mass Spectrometry analysis have been so far unsuccessful.

Are TFEB and MITF able to induce TFE3-L-regulated genes? Could the TFE3-L N-terminal region promote the expression of TFE3-L-specific genes when fused to TFEB? In the discussion, the authors hypothesize that "TFE3-L might indirectly affect expression of specific MITF targets". This hypothesis can be tested using MITF-KO cells.

As requested by the reviewer, we used siRNAs to deplete MITF in ARPE-19 cells and assessed the TFE3-L induced expression of specific target genes. Unfortunately, the results were inconsistent. Whereas we did observe a reduction in the upregulation of PMEL induced by TFE3-L, the levels of other MITF targets were unchanged or even slightly increased (Figure C attached to this rebuttal letter). A recent study has reported the enormous complexity of the relative contribution of different members of the MIT/TFE family to the expression of target genes (Dias et al., 2024). In many cases, is not just the presence or absence of a particular member, but their relative expression level, what determines induction or repression of a particular gene. It is, therefore, unclear, whether our experimental conditions are optimal to assess the MITF contribution to TFE3-L selectivity.

6. Physiological role of TFE3-L. While the study suggests that TFE3-L accumulation promotes migratory and invasive behavior, this isoform also accumulates in non-transformed cells under stress. What is the physiological relevance of TFE3-L expression in these contexts?

This is a very interesting question that we plan to address in future studies. At this point we speculate that TFE3-L may mediate not just increased migration but a profound cytoskeletal reorganization that may be important to facilitate adaptation to stress (for example by increasing nutrient uptake by macropinocytosis under conditions of prolonged starvation or allowing activation of mechanical stress pathways).

7. Literature citation. The manuscript should provide more comprehensive citation of foundational and recent literature. When TFEB/TFE3 regulation of lysosomal biogenesis and autophagy is introduced in the manuscript, the original studies in which these findings were first reported should be cited (PMID: 19556463, 21617040). Likewise, when introducing TFEB/TFE3 regulation by mTORC1, recent publications elucidating the non-canonical regulation of TFEB and TFE3 via the FLCN complex should also be cited (PMID: 32612235, 35358174, 32989250).

The indicated references have been included in the revised version of the manuscript.

Referee #3:

This article conducts a systematic exploration of the partial redundancy and functional disparities of the two isoforms of TFE3. The authors showed that compared with TFE3-S, TFE3-L is associated with metastatic behaviors, which may hold significant relevance in the context of cancer. The key findings of this study are both interesting and significant. It utilizes a variety of molecular biology approaches and provides compelling evidence, thus facilitating a deeper understanding and broadening the functions of TFE3.

We thank the reviewer for the support

However, there are also some inconsistent data within this manuscript. In addition, regarding the effects of TFE3-L on cell migration and invasion compared to TFE3-S, the authors did not explore the underlying mechanisms. It remains to be explored whether such function is related to the autophagy or transcriptional function of TFE3-L. This would contribute to a more profound understanding of the non-canonical functions of TFE3-L and enhance the impact of this article. The following are several suggestions for revisions.

Please note that we have now addressed the contribution of autophagy and the transcriptional activity of TFE3-L to cell-cell adhesion and invasion (see below). In addition, our study shows the superior ability of TFE3-L to induce expression of multiple genes implicated in migration, metastasis, and inhibition of canonical Wnt pathway (Figure 5 and Figure EV5E).

1. In Figure EV1B, subsequent to the EBSS treatment, the level of TFE3-S gradually declined. However, in Figure 1A, TFE3-S remains stable following EBSS starvation. Is this attributable

to the different cell types employed? Could the authors repeat this experiment or provide an explanation for such discrepancies?

We have now quantified the TFE3-S/GAPDH ratio in HEK293T cells incubated with EBSS for different times, corresponding to three independent experiments. As seen in Figure D (attached to this rebuttal letter), the TFE3-S levels only show a slightly significant decrease after 24h. While ARPE-19 cells seem to be very resistant to prolonged starvation, HEK293T are less so. We think that decreased viability of HEK293T at 24h ESBB treatment may explain this minor discrepancy.

2. In Figure 1E, the degradation of TFE3-L by MG132 was not prevented. Instead, its degradation was merely delayed when compared to that under the Re-fed condition. Notably, after 6 hours in particular, the level of TFE3-L still decreased significantly.

Please note that the TFE3-L/TFE3-S ratio remains unchanged in the presence of MG132 (Figure 1F), in contrast with cells that are re-fed in the absence of the inhibitor (Figure 1D). The slight reduction in TFE3-L and TFE3-S levels at 6h MG132 treatment may once again reflect a decrease in cell viability after this harsh treatment conditions (6h proteasome inhibition following 24h starvation). In fact, the smear observed at 6h MG132 seems consistent with increased TFE3-L and TFE3-S cleavage or degradation.

3. Compared to Figure 1C, in Figure 1E why TFE3-S becomes unstable following the addition of MG132?

Please see our response to point 2.

4. In Figure 2F, could you please explain the reason why MLN4924, which is supposed to stabilize TFE3-L, reduces the level of TFE3-M1A? But it seems to stabilize both TFE3-L and TFE3-S in Figure 1J.

Please note that the TFE3-M1A lacks the N-terminal region containing the phosphodegron. As such, the TFE3-M1A mutant is not stabilized by MLN4924 and is not recognized by TFE3 antibody directed against the N-terminal region (Figure 2F). To further confirm these results, we repeated three more times the indicated experiment and performed quantification. As shown in Figure E (attached to this rebuttal letter), the levels of TFE3-M1A do not significantly change in response to MLN4924.

5. In Figure 3E, to which specific WIPI does the term "WIPI" refer? WIPI2?

We specifically analyzed WIPI1. This information has now been included in Figure 3E.

6. In Figure 3E, under basal conditions, overexpression of TFE3-S shows better effects on the expression of autophagy-related genes as compared to TFE3-L. However, in Figure EV3B, subsequent to treatment with EBSS or Torin1, TFE3-L demonstrates a more pronounced

effect in regulating the expression of autophagy-related genes compared to TFE3-S. Under EBSS starvation conditions, both TFE3-L and TFE3-S translocate to the nucleus to promote the expression of the target genes and the protein level of TFE3-S appears to be much higher than that of TFE3-L. Why would TFE3-L have better effects than TFE3-S on autophagy-related genes expression? This discrepancy should also be discussed in the discussion section.

We do not believe that our data support the claim that TFE3-L is more efficient inducing expression of autophagy genes. Under basal conditions, over-expression of both Ad-TFE3-L and Ad-TFE3-S increases expression of autophagic genes efficiently. The reviewer is correct in that in these experiments, Ad-TFE3-S causes higher upregulation of UVRAG, WIPI1, and ATG4A; however, the induction of other genes, such as GABARAPL1, MAP1LC3B, and WDR81 is comparable between both isoforms (Figure 3E). The setting of the experiments shown in Figure EV3B is different, here we show expression of plasmids encoding TFE3-L and TFE3-M1 to low levels (to prevent nuclear accumulation in basal conditions), followed by stress (to induce activation). In this case we just tested one autophagic gene, UVRAG, and the differences in induction of this gene under EBSS and Torin-1 conditions in TFE3-L- and TFE3-M1A-expressing cells were not statically significant.

7. In Figure 4-6, the author investigated the impacts of TFE3-L and TFE3-S on cell proliferation, cell migration, and invasion. TFE3-L demonstrated a much better effect on cell migration and invasion compared with TFE3-S. Does TFE3-L have an impact on the aforementioned phenotypes in the cells with defective autophagy?

This is an interesting suggestion, as several studies have previously described that autophagy-mediated degradation of focal adhesions has a profound effect on cell migration and invasion (Kenific et al., 2016; Bressan et al., 2020; Lu et al., 2021). To address this point, we assessed the effect of TFE3-L over-expression in control and ATG7-depleted cells. Efficient ATG7 depletion, as well as comparable TFE3-L expression, was monitored by qPCR and immunoblot (Appendix Figure S7A-C). We found that the ability of TFE3-L to significantly reduce γ -catenin expression and increase invasion was not affected by depletion of ATG7 (Appendix Figure S7D-G), suggesting that the ability of TFE3-L to alter cell-cell adhesion and migration is not mediated by its ability to induce autophagy.

Are these functions associated with the transcriptional activity of TFE3-L? Is it possible to test the impact of a TFE3-L mutant, which is unable to translocate into the nucleus, on cell migration and invasion?

As requested by the reviewer, we have now mutated the nuclear import signal in TFE3-L. As seen in Appendix Figure S8A, the TFE3-L- Δ NLS mutant was unable to translocate to the nucleus both in control and Torin1-treated conditions. Interesting, and in contrast with TFE3-L, the TFE3-L- Δ NLS mutant failed to increase cell invasion (Appendix Figure S8B,C), suggesting that TFE3-L transcriptional activity is indeed required.

8. In Figure 6, the authors showed that TFE3-L promoted cell migration and invasion of TSC KO cells. However, this experiment was mainly conducted in HeLa cells. Will there be consistent effects in other tumor cells upon TSC knockout or knockdown?

Unfortunately, we do not have any other TSC2-KO cellular model available in the laboratory. However, we further corroborated our observations by showing that TFE3-L is more efficient than TFE3-S in promoting migration in HeLa cells (Appendix Figure S6). Therefore, our results seem to be consistent and reproducible in a variety of cell types.

9. Given that this study is primarily conducted in cultured cells rather than through in vivo investigations, the statement regarding cancer should be made with prudence. Moreover, such limitations should be included in the discussion.

A statement regarding the limitations of the current study has now been included in the discussion.

We want to thank all the reviewers for their constructive and valuable comments. All the points raised were quite useful and helped us to improve our manuscript.

REFERENCES

Bressan C, Pecora A, Gagnon D, Snapyan M, Labrecque S, De Koninck P, Parent M, Saghatelian A. The dynamic interplay between ATP/ADP levels and autophagy sustain neuronal migration in vivo. *Elife*. 2020 Sep 28;9:e56006. doi: 10.7554/eLife.56006. PMID: 32985978; PMCID: PMC7556871.

Dias D, Oliveira E, Martí-Díaz R, Andrews S, Chocarro-Calvo A, Bellini A, Mosteo L, Vivas García Y, Chauhan J, Li L, García-Martínez JM, Rodríguez-López JN, Maria-Engler SS, García-Jiménez C, Sanchez-Del-Campo L, Louphrasitthiphol P, Goding CR. Functional specialization of MITF, TFEB and TFE3 drives radically distinct adaptive gene expression programs in melanoma. *bioRxiv* [Preprint]. 2024 Dec 23:2024.12.23.629393. doi: 10.1101/2024.12.23.629393. PMID: 39764020; PMCID: PMC11703276.

Gupta S, McCarthy MR, Tjota MY, Antic T, Cheville JC. Metastatic renal cell carcinoma with fibromyxomatous stroma associated with tuberous sclerosis or MTOR, TSC1/TSC2-Mutations: A Series of 4 cases and a review of the literature. *Hum Pathol*. 2024 Nov;153:105680. doi: 10.1016/j.humpath.2024.105680. Epub 2024 Nov 8. PMID: 39522702.

Kenific CM, Wittmann T, Debnath J. Autophagy in adhesion and migration. *J Cell Sci.* 2016 Oct 15;129(20):3685-3693. doi: 10.1242/jcs.188490. Epub 2016 Sep 26. PMID: 27672021; PMCID: PMC5087656.

Lu J, Linares B, Xu Z, Rui YN. Mechanisms of FA-Phagy, a New Form of Selective Autophagy/Organellophagy. *Front Cell Dev Biol.* 2021 Dec 7;9:799123. doi: 10.3389/fcell.2021.799123. PMID: 34950664; PMCID: PMC8689057.

Martina JA, Puertollano R. Rag GTPases mediate amino acid-dependent recruitment of TFEB and MITF to lysosomes. *J Cell Biol.* 2013 Feb 18;200(4):475-91. doi: 10.1083/jcb.201209135. Epub 2013 Feb 11. PMID: 23401004; PMCID: PMC3575543.

Martina JA, Diab HI, Lishu L, Jeong-A L, Patange S, Raben N, Puertollano R. The nutrient-responsive transcription factor TFE3 promotes autophagy, lysosomal biogenesis, and clearance of cellular debris. *Sci Signal.* 2014 Jan 21;7(309):ra9. doi: 10.1126/scisignal.2004754. PMID: 24448649; PMCID: PMC4696865.

Martina JA, Guerrero-Gómez D, Gómez-Orte E, Antonio Bárcena J, Cabello J, Miranda-Vizuete A, Puertollano R. A conserved cysteine-based redox mechanism sustains TFEB/HLH-30 activity under persistent stress. *EMBO J.* 2021 Feb 1;40(3):e105793. doi: 10.15252/embj.2020105793. Epub 2020 Dec 14. PMID: 33314217; PMCID: PMC7849306.

Meredith L, Chao T, Nevler A, Basu Mallick A, Singla RK, McCue PA, Bowne WB, Jiang W. A rare metastatic mesenteric malignant PEComa with TSC2 mutation treated with palliative surgical resection and nab-sirolimus: a case report. *Diagn Pathol.* 2023 Apr 11;18(1):45. doi: 10.1186/s13000-023-01323-x. PMID: 37041531; PMCID: PMC10088294.

Tapia PJ, Martina JA, Contreras PS, Prashar A, Jeong E, De Nardo D, Puertollano R. TFEB and TFE3 regulate STING1-dependent immune responses by controlling type I interferon signaling. *Autophagy.* 2025 Sep;21(9):2028-2045. doi: 10.1080/15548627.2025.2487036. Epub 2025 Apr 20. PMID: 40195022; PMCID: PMC12363505.

Figure A

(A) Schematic diagram of the cDNA structure of TFE3-L and TFE3-S representing exons 1 through 10 and exons 3 through 10 respectively, their corresponding 5'UTR and the position of the start and stop codons. Exon sizes are not to scaled, and boundaries are approximate. **(B)** Gel electrophoresis analysis showing the RT-PCR amplified cDNA fragments of TFE3. Black and orange arrows represent common reverse primer targeting the stop region of both isoforms, blue and green arrows represent forward primers that target the 5'UTR of TFE3-L and TFE3-S respectively. **(C)** Complete sequence chromatogram of the subcloned cDNA amplified fragments (a) and (d) shown in (B). Colored arrow indicate the beginning and end of each cDNA fragment as described in (B), and the start codons corresponding to methionines 1 and 106 are indicated with red rectangles.

A**B**
Epigenetic profile of human *TFE3* gene. Genomic region encompassing the first four exons of the human *TFE3* gene visualized using the UCSC genome (A) or the Cistrome DB WashU browsers (B). (A) T2T Encode Track was used to check the epigenetic profiles of the genomic region. Black bars show histone methylation (H3K4me3 and H3K4me1) and acetylation (H3K27ac) profiles in different cell line datasets. Red box highlights a H3K4me3 peak in HAP-1 cell dataset right upstream of exon 3. (B) For the same genomic region, Chip-seq polR2a-pS5 profiles from all the available datasets are shown. Peaks in proximity of exon1 and upstream the exon 3 are highlighted in light blue.

Figure B

A**B**
(A) Representative western blot showing MITF, MYC and GAPDH protein levels from ARPE19 cells transfected with a siRNA control and a siRNA against MITF for 72h. Cells were infected with Ad-Null, Ad-rTFE3-L or Ad-rTFE3-S for 24h. (B) Relative quantitative RT-PCR analysis of the mRNA expression of different genes (*PMEL*, *ATP6V1C1*, *TBX6* and *HEY2*). Data are presented as mean \pm SD. (ns) not significant, (*) $p < 0.05$, (**) $p < 0.01$, (***) $p < 0.001$, and (****) $p < 0.0001$ (One-way ANOVA followed by Dunnett's multiple comparison post-test, $n=3$).

Figure C

Quantification of protein levels showing TFE3-S/GAPDH ratio expressed as fold change corresponding to the immunoblot analysis from Figure EV1 showing protein lysates of HEK293T cells treated with EBSS for the indicated times. Data are presented as mean \pm SD. (ns) not significant, (*) $p < 0.05$. (One-way ANOVA followed by Tukey's multiple comparison post-test).

Figure D

A**B**
Representative western blot (A) and quantification (B) of protein levels showing TFE3-MYC/GAPDH ratio expressed as fold change from ARPE19 cells transfected with rTFE3-L and rTFE3-M1A for 24h and treated with MLN4924 1 μ M for 6h. Data are presented as mean \pm SD. (ns) not significant (One-way ANOVA followed by Dunnett's multiple comparison post-test, n=3). Asterisk (*) indicate unspecific band.

Figure E

Dear Dr. Puertollano

Thank you for the submission of your revised manuscript to EMBO reports. As you know, we have meanwhile received the reports from all three referees who support publication pending a few minor changes that I kindly ask you to address.

From the editorial side, there are also a few things that we need before we can proceed with the official acceptance of your study.

Please provide a point-by-point response to both, the referee concerns and the editorial points listed below.

EDITORIAL POINTS:

- Please provide up to 5 keywords on the title page.
- Please rename the Competing Interests section to Disclosure and Competing Interests Statement.
- Regarding the Author Contributions, we now use CRedit to specify the contributions of each author in the journal submission system. Therefore, please remove the Author Contributions from the manuscript file and make sure that the author contributions in our online manuscript tracking system are correct and up-to-date. The information you specified in the system will be automatically retrieved and typeset into the article. You can enter additional information in the free text box provided, if you wish. See also our guide to authors <https://www.embopress.org/page/journal/14693178/authorguide#authorshipguidelines>.
- Please provide the EV figures as separate production quality figure files. Their legends should stay in the manuscript at the end with the header "Expanded View Figure Legends".
- Table EV1 and Table EV2 have a lot of data and should be datasets. Please update them to Dataset EV1 and Dataset EV2 in all places (source file names, legends, titles in the system, callouts in the manuscript).
- Table EV3 should be uploaded as a separate Expanded View Content file titled Table EV1. Please update the name and callouts in all places (source file names, legends, titles in the system, callouts in the manuscript, the Author Checklist and the Reagents and Tools table).
- The legends of EV tables and Datasets need to be provided only in their respective files (not in the manuscript).
- Each movie should be zipped up with its legend, which is provided as a separate README.text file. This produces one zip folder per movie, which is uploaded. The movie legends should be removed from the manuscript.
- Appendix: please add page numbers to the table of content on the first page.
- Please define the number and nature of all replicates and the exact p-values in all Appendix Figure legends.
- Please rename the section "Data and code availability" to "Data availability" and move it before the Acknowledgments. Since section should only refer to datasets deposited in public repositories, I kindly ask you to remove the following statements:
 - This paper does not report original code.
 - Any additional information required to reanalyze the data reported in this paper is available from the lead contact upon request.
- Please provide the exact p values in the legends of figures 1B, K; 3E, 4E, G; 5B, D, F, H; 6B, E; EV1 C, E, G; EV3 A, B; EV4 C, D, F, H, J; EV5 A, C, E, unless the p-value is <0.0001.
- Finally, EMBO Reports papers are accompanied online by
 - A) a short (1-2 sentences) summary of the findings and their significance,
 - B) 2-3 bullet points highlighting key results and
 - C) a schematic summary figure that provides a sketch of the major findings (not a data image).Please provide the summary figure as a separate file in PNG or JPG format at a size of 550x300-600 pixels (width x height). Please note that the size is rather small and that text needs to be readable at the final size. Please send us this information along with the revised manuscript.

Kind regards,

=====

Referee #1:

All questions and concerns have been thoroughly addressed in this revision.

Referee #2:

Although the revised version of the manuscript has improved, an important issue that remains unresolved concerns the regulation of TFE3-L by mTORC1 (point 3 from my previous review).

While the authors claim that TFE3-L levels are regulated by mTORC1, treatment with torin1, a potent mTOR inhibitor, appears to have only a minor effect, if any, on TFE3-L stabilization (Figure 1L and Appendix Figure 2B). A point of concern is the way TFE3-L quantification has been performed throughout the manuscript, expressed as the ratio TFE3-L/TFE3-S. This approach is not accurate, as torin treatment markedly reduces TFE3-S levels, leading to confounding results. Therefore, the authors should normalize TFE3-L levels to GAPDH (TFE3-L/GAPDH). Nevertheless, other stressors that promote TFE3 nuclear translocation without impairing 4EBP1 phosphorylation, such as LLOMe, NaAsO₂, and chloroquine, potently promote TFE3-L accumulation. These findings suggest that non-canonical mTORC1 signaling, rather than the canonical mTORC1 pathway, is involved in the regulation of TFE3-L. Thus, the authors' conclusions do not seem to be supported by the data and should be revised accordingly.

Additionally, while the authors acknowledge in their rebuttal that "other stressors, such as LLOMe, are known to prevent non-canonical mTOR signaling," there is no mention of such TFEB/TFE3 regulatory mechanism in the introduction and discussion sections. I recommend explicitly incorporating this concept and ensuring that references are used accurately and consistently, as some of them appear misplaced. For example, Cui et al. 2023 did not demonstrate that TFEB and TFE3 need to be recruited to the lysosomal limiting membrane (page 3). Similarly, Li et al. 2022, Nakamura et al. 2020, and Napolitano et al. 2020 are not the primary references demonstrating mTORC1-dependent TFEB/TFE3 phosphorylation. Rather, Cui, Napolitano, and Li et al. established that TFEB and TFE3 are recruited by mTORC1 through an FLCN-regulated mechanism involving GDP loading of RagC, while Nakamura et al. showed that LLOMe and lysosomal damage specifically affect this non-canonical mTORC1 pathway. These inaccuracies should be corrected, and the discrepancy between torin treatment and other stressors should be discussed.

Finally, it remains unclear why, in response to point 3 of my previous revision, the authors were unable to perform high-efficiency transfections of active/inactive Rag GTPases, as in the original submission expression of active/inactive Rag GTPases was shown to efficiently alter TFE3-L levels (Figure EV1D and EV1G). To confirm that this effect is mTORC1-dependent, it would have been sufficient to include torin and/or EBSS treatments under the same experimental conditions.

Referee #3:

The authors have carefully and comprehensively addressed all the concerns.

Rebuttal Letter

Referee #1:

All questions and concerns have been thoroughly addressed in this revision.

Referee #2:

Although the revised version of the manuscript has improved, an important issue that remains unresolved concerns the regulation of TFE3-L by mTORC1 (point 3 from my previous review). While the authors claim that TFE3-L levels are regulated by mTORC1, treatment with torin1, a potent mTOR inhibitor, appears to have only a minor effect, if any, on TFE3-L stabilization (Figure 1L and Appendix Figure 2B). Nevertheless, other stressors that promote TFE3 nuclear translocation without impairing 4EBP1 phosphorylation, such as LLOMe, NaAsO₂, and chloroquine, potently promote TFE3-L accumulation. These findings suggest that non-canonical mTORC1 signaling, rather than the canonical mTORC1 pathway, is involved in the regulation of TFE3-L. Thus, the authors' conclusions do not seem to be supported by the data and should be revised accordingly.

Please note that the mTORC1-dependent phosphorylation of TFE3-S47 was suggested by a previous study (Nardone et al., 2023). We believe that our results are consistent with this idea, as we show that any stimuli that prevents mTORC1-dependent TFE3 phosphorylation, either through direct mTORC1 inhibition or by preventing the interaction of TFE3 with Rags, causes TFE3-L stabilization overtime. However, the regulation of the TFE3 phosphodegron is not the main focus of our current study, and we cannot rule out additional intricacies in the regulation of TFE3 stability. For example, even when the mTORC1-dependent phosphorylation of TFE3 is completely inhibited (see for example the lack of TFE3 S-321 phosphorylation in cells treated with NaAsO₂ shown in Figure 1L) it still takes several hours to significantly rise the levels of TFE3-L (Appendix Figure S4B and C). It is therefore possible that additional kinases or regulatory mechanisms play a role. Alternatively, our results may just indicate that it takes some time to accumulate enough TFE3-L to allow detection by the anti-TFE3 antibody used in these experiments.

A point of concern is the way TFE3-L quantification has been performed throughout the manuscript, expressed as the ratio TFE3-L/TFE3-S. This approach is not accurate, as torin treatment markedly reduces TFE3-S levels, leading to confounding results. Therefore, the authors should normalize TFE3-L levels to GAPDH (TFE3-L/GAPDH).

We agree with the reviewer in that prolonged (over 12h) Torin-1 treatment reduces TFE3-S levels, an effect that may be due to an overall inhibition in global protein synthesis. However, even at these prolonged treatment times, the levels of TFE3-L are significantly higher than in control conditions (Appendix Figure S4C). Also, please note that Torin-1 almost completely blocked TFE3-L degradation under refed conditions (Appendix Figure S2B-C).

We would prefer keeping the TFE3-L/TFE3-S quantification currently shown in the manuscript, as it provides important information about the relative abundance of each one of the two TFE3 isoforms. However, in response to the reviewer's request, we now included quantification of the TFE3-L/GAPDH ratio for some of our key experiments (Appendix

Figure S1A and Figure EV1D). This new quantification did not change the conclusions of our study.

Additionally, while the authors acknowledge in their rebuttal that "other stressors, such as LLOMe, are known to prevent non-canonical mTOR signaling," there is no mention of such TFEB/TFE3 regulatory mechanism in the introduction and discussion sections. I recommend explicitly incorporating this concept and ensuring that references are used accurately and consistently, as some of them appear misplaced. For example, Cui et al. 2023 did not demonstrate that TFEB and TFE3 need to be recruited to the lysosomal limiting membrane (page 3). Similarly, Li et al. 2022, Nakamura et al. 2020, and Napolitano et al. 2020 are not the primary references demonstrating mTORC1-dependent TFEB/TFE3 phosphorylation. Rather, Cui, Napolitano, and Li et al. established that TFEB and TFE3 are recruited by mTORC1 through an FLCN-regulated mechanism involving GDP loading of RagC, while Nakamura et al. showed that LLOMe and lysosomal damage specifically affect this non-canonical mTORC1 pathway. These inaccuracies should be corrected, and the discrepancy between torin treatment and other stressors should be discussed.

As requested by the reviewer, we have now expanded the introduction to discuss the role of FLCN in TFEB/TFE3 regulation and have added the corresponding references.

Finally, it remains unclear why, in response to point 3 of my previous revision, the authors were unable to perform high-efficiency transfections of active/inactive Rag GTPases, as in the original submission expression of active/inactive Rag GTPases was shown to efficiently alter TFE3-L levels (Figure EV1D and EV1G). To confirm that this effect is mTORC1-dependent, it would have been sufficient to include torin and/or EBSS treatments under the same experimental conditions.

We did perform the experiments suggested by the reviewer, as we indicated in our previous rebuttal letter. However, while expression of inactive Rags was sufficient to cause at least a partial inactivation of mTORC1 (see phospho-4EBP1 levels in Figure EV1E), resulting in accumulation of TFE3-L, expression of active Rags was unable to block mTORC1 inactivation under conditions of prolonged EBSS treatment (see Figure A attached to this rebuttal letter). We hypothesize that this may be due to insufficient level of expression of active Rags or low transfection efficiency. It is also possible that under prolonged starvation conditions other mechanisms (e.g. Rheb inactivation due to the lack of growth factors in the EBSS medium) cause Rag-independent mTORC1 inactivation. In any case, without an efficient inhibition of mTORC1 inactivation, we are unable to assess TFE3-L stabilization.

Referee #3:

The authors have carefully and comprehensively addressed all the concerns.

We thank all the reviewers for their constructive and valuable comments. All the points raised were quite useful and helped us to improve our manuscript.

Figure A. Expression of TFE3 isoforms in response to stress. Immunoblot analysis of protein lysates from ARPE19 cells mock transfected or expressing either active Rag B/D or Rag A/C heterodimers treated with EBSS or Torin-1 for 8h.

Dr. Rosa Puertollano
NIH
Laboratory of Cell Biology, National Heart, Lung, and Blood Institute
50 South Drive
Building 50, room 3537
Bethesda 20892
United States

Dear Dr. Puertollano,

I am very pleased to accept your manuscript for publication in the next available issue of EMBO reports. Thank you for your contribution to our journal.

Yours sincerely,
